# Whey Valorization in Functional Jellies: A Nutritional and Technological Approach

**DOI:** 10.3390/foods14183193

**Published:** 2025-09-13

**Authors:** Diana Fluerasu (Bălțatu), Monica Negrea, Christine Neagu, Sylvestre Dossa, Călin Jianu, Dacian Lalescu, Adina Berbecea, Liliana Cseh, Ileana Cocan, Corina Misca, Mariana Suba, Vlad Muresan, Anda Tanislav, Ersilia Alexa

**Affiliations:** 1Faculty of Food Engineering, University of Life Sciences “King Mihai I” from Timisoara, Aradului Street No. 119, 300645 Timisoara, Romania; fluerasu.diana@usvt.ro (D.F.); christine.neagu@usvt.ro (C.N.); sylvestredossa04@gmail.com (S.D.); calinjianu@usvt.ro (C.J.); lalescu@usvt.ro (D.L.); ileanacocan@usvt.ro (I.C.); corinamisca@usvt.ro (C.M.); ersiliaalexa@usvt.ro (E.A.); 2“Food Science” Research Center, University of Life Sciences “King Mihai I” from Timisoara, Aradului Street No. 119, 300645 Timisoara, Romania; 3Faculty of Agriculture, University of Life Sciences “King Mihai I” from Timisoara, Aradului Street No. 119, 300645 Timisoara, Romania; adina_berbecea@usvt.ro; 4Romanian Academy, “Coriolan Dragulescu” Institute of Chemistry, Mihai Viteazu No. 24, 300223 Timisoara, Romania; marianasuba@gmail.com; 5Faculty of Food Science and Technology, University of Agricultural Sciences and Veterinary Medicine Cluj-Napoca, 3-5 Manastur St., 400372 Cluj-Napoca, Romania; vlad.muresan@usamvcluj.ro (V.M.); anda.tanislav@usamvcluj.ro (A.T.)

**Keywords:** berries, functional jellies, whey, FTIR, syneresis, phytochemical, color analysis, texture

## Abstract

The purpose of this paper is to evaluate the nutritional, functional, and technological potential of whey resulting as a by-product in the dairy industry, as such or mixed with berries (blueberries, strawberries, and raspberries) to obtain healthy jellies with added value. In this regard, the following parameters were analyzed: protein content, total amino acids, total mineral substances, macro- and microelements, antioxidant capacity, and total polyphenols. Also, the storage stability, textural and color parameters, FTIR spectra, and microstructures of jellies were analyzed. The results obtained showed that the protein content ranged from 4.18% to 4.51%, with a general increase observed in the variants with added whey and berries. Regarding total mineral substances, a significant increase was noted in jellies with added whey (0.34%) and strawberries (0.35%), compared to the control (0.15%). Whey jellies presented the highest levels of K, Ca, Mg, Zn, and Fe, while samples with added fruits completed the microelement (Mn, Cu, Ni, and Cr) content. The storage stability at 4 °C and the evolution of pH and acidity confirm that the products maintain their structure, while when stored at ambient temperature an acceleration of the decrease in pH and an increase in acidity are observed after 14 days. The jellies with combined additions (whey and berries) presented the most favorable microstructure, which supports the use of synergistic functional ingredients in the development of innovative products with high nutritional and sensory value. The FTIR spectra reflect the composition of the ingredients used. Based on obtained results, it can be concluded that whey represents a versatile and sustainable resource for obtaining functional jellies, offering both nutritional benefits and favorable economic and ecological perspectives.

## 1. Introduction

Whey, a significant by-product of the dairy industry, has evolved from a low-value waste material into a strategic source of bioactive compounds with functional roles in modern foods. Its high-quality protein content (including β-lactoglobulin, α-lactalbumin, and immunoglobulins), along with lactose, vitamins, and minerals, makes it suitable for applications in the development of jellies with enhanced nutritional functionality [1]. The incorporation of whey into functional jelly formulations can improve nutritional value by increasing the content of high-biological-value proteins and enhancing antioxidant capacity. Moreover, the bioactive compounds in whey may help reduce oxidative stress, support immune function, and regulate the intestinal microbiota [2]. The addition enables the creation of high-protein-density jellies, with the potential to promote muscle health, strengthen immunity, and modulate gut microbiota [3]. Furthermore, bioactive peptides generated through the enzymatic hydrolysis of whey proteins may exert antioxidant, antihypertensive, and anti-inflammatory effects, thereby reinforcing the functional character of the final product [4]. From a technological perspective, whey proteins play an essential role in the structure and stability of gels [5]. They can act synergistically with gelling agents (such as gelatin, pectin, or carrageenan) to form stable networks with the desired texture and consistency. These proteins also enhance water retention capacity and promote the uniform distribution of other functional ingredients, including probiotics or plant extracts [6,7].

Recent trends in the formulation of functional jellies include fortification with natural antioxidants derived from fruits, medicinal plants, or isolated polyphenols. When combined with whey proteins, these bioactive compounds can enhance the antioxidant potential and improve product stability during storage [8].

Beyond their functional and technological advantages, the valorization of whey contributes to reducing the environmental footprint of the dairy industry, supporting the transition toward a circular economy—an essential direction in line with European and global sustainable development goals [9,10].

The recovery and use of whey in the form of functional jellies help reduce agri-food waste while promoting circular economy practices. Transforming this by-product into a value-added food aligns with the principles of sustainable development and the “zero waste” strategy [11,12].

Whey thus represents a versatile and sustainable ingredient for obtaining functional jellies, offering both nutritional benefits and favorable economic and ecological implications. Its integration into innovative products responds to current consumer demands for healthy, natural, and functional foods.

In this context, the present paper investigates the potential use of whey from the dairy industry in the production of value-added functional jellies enriched with bioactive compounds from berries (blueberries, strawberries, and raspberries). The products were characterized in terms of physical–chemical, nutritional, textural, storage stability, and phytochemical properties. This paper addresses the need for innovation in the food industry within the framework of the circular and sustainable economy.

## 2. Materials and Methods

### 2.1. Materials

For the preparation of the jellies, the following ingredients were used: cow’s sweet whey derived from cheese production, gelatin (Dr. Oetker, Bielefeld, Germany), beetroot sugar Diamant (Oradea, Romania), water, fruit juices (strawberries, blueberries, and raspberries). The fruits were purchased from local markets, while the whey was sourced from BRO-LACT FARM, a dairy factory in Mehedinți district, Romania. To obtain 300 mL of juice, 700–800 g of fresh fruit was used, which was cleaned, washed, and blended using a mixer (Braun TriForce Power Blend 9, Braun, Neu-Isenburg, Germany), after which the resulting product was strained to obtain the liquid sample.

### 2.2. Jelly Preparation

The jelly formulations and recipes are presented in Table 1. Gelatin (10 g) was hydrated in 50 mL of cold water for 10 min. The fruit juice, strawberry, raspberry, or blueberry (300 mL), was combined with sugar (80 g) and heated to 60–70 °C without reaching the boiling point. The hydrated gelatin was added to the warm fruit juice to obtain a uniform mixture, which was cooled to around 40 °C. Cow’s milk whey (200 mL), previously filtered, was gently incorporated into this mixture. The final preparation was poured into molds or glass containers and refrigerated for at least 4 h to ensure complete gelation. The obtained samples are presented in the Figure 1.

### 2.3. Determination of Physical–Chemical Parameters of Jellies

#### 2.3.1. Determination of pH

The pH of the jellies was measured with a digital pH meter (inoLab pH 720 pH meter (Xylem Analytics, Weilheim, Germany)). For this, 10 g of each jelly and 90 mL water were homogenized for 60 s and stood for 3–5 min to release air bubbles, and then the pH was measured [13]. Measurements were performed initially and after 7 and 14 days of storage at 2 different temperatures (4 and 20–25 °C) to assess jelly stability. All determinations were conducted in triplicate.

#### 2.3.2. Determination of Titrable Acidity (TA)

To determine the titratable acidity, 10 g of jelly was homogenized with 50 mL of distilled water to ensure the complete dissolution of the soluble components. Subsequently, titration was performed with a 0.1 N sodium hydroxide (NaOH) solution, using phenolphthalein as a pH indicator. The titration was performed until a pale pink color appears, indicating that the equivalence point had been reached [14]. The volume of NaOH consumed is then used to calculate the titratable acidity expressed as a percentage of lactic acid. Each sample was analyzed in triplicate. These parameters were determined initially and after 7 and 14 days, at 2 different temperatures (4 and 20–25 °C) in order to evaluate the stability of jellies.(1)$$TA\;\left(\%\right)=\frac{V\times N\times f\times 100}{m}$$
where the following was the case:
V = volume of NaOH solution used in titration (mL).N = normality of NaOH solution (mol/L equivalents).f = conversion factor, specific to the reference acid (g acid/mmol equivalent).m = mass of the analyzed sample (g).

#### 2.3.3. Determination of Water Content

The moisture content of the jellies was determined by the thermogravimetric method, using a thermobalance (MetroToledo, Columbus, OH, USA). Representative samples (approximately 2–5 g) were placed on the support of the apparatus and subjected to a controlled heating process, at a temperature of 100–105 °C, until a constant mass was obtained. During drying, the weight loss was automatically recorded by the equipment, which corresponded to the evaporation of water from the sample. The moisture content was calculated based on the difference between the initial and final mass of the samples, related to the initial mass.

#### 2.3.4. Protein and Total Amino Acid Determination

Protein determination was performed using the Kjeldahl method according to ISO 8968-1|IDF 20-1:2014 [15]. Total free amino acids were determined using the spectrophotometric method [16]. For the determination of total free amino acids, 1 g of sample was weighed and mixed with 24 mL of hydroalcoholic solution (70:30 = ethanol:H_2_O). The mixtures were homogenized for 30 min using a DLAB shaker (SK-L330-PRO, China) and then filtered. Subsequently, 1 mL of the filtrate was transferred into test tubes, to which 0.5 mL of phosphate buffer solution (pH 8.04; 1/15 mol/L) and 0.5 mL of ninhydrin solution (2% + 0.8 mg/mL SnCl_2_·2H_2_O) were added. The samples were shaken for 30 min, then incubated in an oven (BINDER GmbH, Tuttlingen, Germany) at 103 ± 4 °C for 10 min. After cooling, the absorbance was measured spectrophotometrically using a UV-Vis spectrophotometer (Specord 205, Analytik Jena AG, Jena, Germany) at 570 nm. All analyses were performed in triplicate.

#### 2.3.5. Determination of Total Mineral Substances

The total mineral content was determined by calcination [17]. Approximately 2–5 g of samples was weighed into pre-dried porcelain crucibles and incinerated in a muffle furnace (190945, Nabertherm, Lilienthal, Germany). The temperature was gradually raised and maintained at 500–550 °C until a constant weight of white ash was obtained. After cooling in a desiccator, the crucibles were weighed, and the mineral content (%) was calculated as the ratio of ash weight to the initial sample weight. All samples were analyzed in triplicate.

### 2.4. Determination of Macro- and Microelements

The content of macro- and microelements was determined using atomic absorption spectroscopy (AAS) with a Varian 220 FAA instrument (Agilent Technologies, Palo Alto, CA, USA), following the method described by Ruja et al. [18]. In brief, the ash sample obtained as described in Section 2.3. was treated with 10 mL of 20% hydrochloric acid (Chimreactiv, Bucharest, Romania). The resulting solutions were analyzed individually by AAS, with each element measured at its specific wavelength using element-specific cathode lamps for K, Ca, Mg, Na, Fe, Zn, Cu, Mn, Ni, and Cr. The instrument was calibrated prior to each analytical series using standard reference solutions. Results were expressed in mg/kg (ppm) of the dry sample. All measurements were performed in triplicate.

### 2.5. Determination of Phytochemical Profile

#### 2.5.1. Determination of the Total Phenolic Content (TPC)

The total phenolic content of jelly extracts was determined using the Folin–Ciocâlteu method [19]. To obtain the alcoholic extract, 1 g of sample was mixed with 10 mL of ethanol 70% and were mixed for 30 min using a Holt mechanical shaker (IDL, Freising, Germany). At 0.5 mL of extract, 1 mL of Folin–Ciocâlteu reagent (diluted 1:10) (Sigma-Aldrich Chemie GmbH, Munich, Germany) and 1 mL of sodium carbonate solution (60 g/L) (Geyer GmbH, Renningen, Germany) were added. The mixture was incubated at 50 °C for 30 min in an INB500 incubator (Memmert GmbH, Schwabach, Germany). Absorbance was then measured at 750 nm using a UV-Vis spectrophotometer, a Specord 205 model (Analytik Jena AG, Jena, Germany). Results were expressed as mg of gallic acid equivalents (GAEs) per 100 g of sample. All analyses were performed in triplicate.

#### 2.5.2. Determination of Antioxidant Activity

The antioxidant activity of jellies was assessed using a 2,2-diphenyl-1-picrylhydrazyl (DPPH) assay (Sigma-Aldrich; Merck KGaA, Darmstadt, Germany) [20]. The reaction mixture consisted of 1 mL of extract (prepared as described in Section 2.5.1.) and 2.5 mL of 0.03 mM DPPH solution. The concentrations of the investigated extracts were 10, 5, 2.5, 1.25, and 0.67 mg/mL. The mixture was incubated for 30 min at room temperature in the dark. After incubation, the samples were filtered to remove solid particles that could cause turbidity, and the absorbance was measured at 518 nm using a Specord 205 UV-Vis spectrophotometer (Analytik Jena AG, Jena, Germany). Ethanol was used as a negative control and ascorbic acid as a positive control. All analyses were performed in triplicate.

### 2.6. Determination of Storage Stability

To evaluate the structural stability of the jellies, a syneresis test was performed over an extended storage period. The samples were stored at a constant temperature of 4 °C and ambient conditions (20–25 °C) and analyzed after 7 and 14 days. At each time point, the gels were visually examined, and any liquid spontaneously released on their surface was carefully separated without disturbing the integrity of the gel’s integrity. The released liquid was collected and accurately weighed, then expressed as a percentage relative to the initial sample mass. Water loss (%) was calculated by comparing the weight of the separated liquid to the total mass of the gel before storage. According to the evaluation criteria, water loss below 5% indicated good stability of the gel network, reflecting effective water retention within the polymeric matrix [21].

### 2.7. Determination of Color Parameters

The color characteristics of jelly products were determined using a colorimeter (KONICA MINOLTA, Tokio, Japan), based on the CIE Lab* color space, internationally standardized for colorimetric evaluations. In this system, the parameter *L** expresses lightness, ranging from 0 (black) to 100 (white), *a** defines the position on the green (negative)–red (positive) axis, while *b** represents the blue (negative)–yellow (positive) axis [22].

In addition to the basic Lab* coordinates, derived parameters were calculated: chroma (*C**), which describes color saturation, and hue angle (*h*°), which indicates the actual color perceived by the human eye.

For analysis, the jelly samples were cut into uniform slices, and measurements were taken at multiple points on the surface in order to obtain representative mean values. All analyses were performed in triplicate.

### 2.8. Texture Profile Analysis

The samples were analyzed using a CT3 Texture Analyzer (Brookfield Engineering Labs, Middleboro, MA, USA) TA44 cylindrical probe (4 mm diameter) mounted on a 10 kg compression cell. Each sample was subjected to a double compression test, with 50% deformation of its original height, at a constant speed of 1 mm/s. The following texture parameters were recorded: hardness (N), adhesiveness (mJ), cohesiveness, springiness index, gumminess (N), and chewiness (mJ). All samples were stored at 4 °C and evaluated directly at this temperature [23]. All analyses were performed in triplicate.

### 2.9. Fourier Transform Infrared Spectroscopy (FTIR)

For FTIR analysis equipment, Nicolet Is50 FT-IR (Thermo Fisher Scientific, Waltham, MA, USA) equipped with an ATR crystal was used. FTIR-ATR spectroscopy, or Attenuated Total Reflectance Fourier Transform Infrared spectroscopy, was performed at room temperature by using a Nicolet™ iS50 FTIR Spectrometer. The IR spectra were obtained in the spectral range of 4000–400 cm^−1^, with 32 scans at a 4 cm^−1^ resolution [24].

### 2.10. Microscopic Evaluation

The microstructure of gels prepared with whey and fruit juice was evaluated using simple optical microscopy to assess homogeneity, phase distribution, and potential aggregations.

A thin jelly slice (~1 mm thick) was cut with a sterile knife or scalpel and placed on a clean microscope slide (Olympus CX-41, Olympus Corporation, Tokyo, Japan). A drop of diluted methylene blue is added to enhance structural contrast. The sample was then covered with a coverslip, ensuring that no air bubbles were trapped. Microscopic observation started at low magnification (100×) to locate areas of interest, followed by higher magnifications (400×) to examine fine structural details of the jelly [25].

### 2.11. Statistical Analysis

All measurements were carried out in triplicate, and the results are reported as the mean ± standard deviation (SD). We repeated all trials on independent fresh batches and analyzed technical replicates per batch. Differences among means were assessed by Duncan’s Multiple Range Test following analysis of variance (ANOVA). Statistical significance was considered at *p* < 0.05. All statistical analyses were performed using R Statistical Software (version 4.3.3; R Core Team, 2023).

## 3. Results and Discussion

### 3.1. Physical–Chemical Parameters of Whey Jellies

The protein, free amino acid, and total mineral contents of the raw materials and jelly samples are presented in Table 2, while the variations in time of humidity (or the temporal variations in moisture), pH values, and titratable acidity are shown in Figure 2.

The CJ sample has the highest moisture content, 70.48 ± 0.34%, compared to 62.34% for WhJ (whey), which signifies an excess of +8.14 percentage points (+13.1% relative) and supports the idea of a softer gel in the absence of whey proteins. The proteins increase from 4.18% (CJ) to 4.30% (WhJ), while the free amino acids (FAAs) rise from 1230.78 ppm to 1423.01 ppm (+13.5%) with the addition of whey in jellies related to the control without whey. The total minerals are similar: 0.346% (CJ) vs. 0.336% (WhJ), a small and statistically insignificant difference. Overall, the transition from water to whey reduces moisture and increases nitrogen content (proteins + FAAs), thus compacting the gel network and enhancing its stability.

The higher moisture of the control jelly (CJ, 70.48%) is consistent with a softer, more open gel network and reduced WHC; in soft hydrogels, greater free water correlates with lower hardness and higher serum release (syneresis). Conversely, whey-based jellies (WhWJ, WhSJ, WhBJ, and WhRJ; 62.3–63.2%) benefit from whey protein (α-lactalbumin, β-lactoglobulin) gelation and protein–polysaccharide interactions, which tighten the network, immobilize water, and reduce syneresis. Studies on acidified milk model gels show that added whey protein ingredients systematically decrease water mobility and syneresis and increase water holding capacity (WHC), with microstructure–water mobility–WHC tightly correlated. Similar improvements in water retention and texture are reported for whey-containing composite gels (including gelatin–globular protein systems) [26,27].

Blueberry jelly (WhBJ) has the highest moisture content by a slight margin (63.22%), possibly due to the higher natural water content of the fruit. Raspberry jelly (WhRJ) and strawberry jelly (WhSJ) are very close, demonstrating good consistency of the technological process and a balanced texture.

The protein content ranged from 4.18% (CJ) to 4.51% (WhSJ), with a general increase observed in the variants with added whey and fruits compared to the control. This result confirms that whey is a rich source of proteins with high biological value, contributing to the increase in the nutritional value of the final products [28]. The WhSJ samples (whey + strawberry jellies) displayed comparatively higher protein levels but lower free amino acid contents than the other formulations. This outcome can be attributed to both compositional and interactional factors. Strawberries contribute relatively modest amounts of free amino acids compared with raspberries or blueberries, while being richer in organic acids and phenolic compounds. These phenolics, particularly ellagic acid derivatives, are prone to interact with whey proteins through non-covalent bonds, thereby stabilizing the protein matrix and limiting proteolysis or the release of small peptides and amino acids. The stronger protein–polyphenol associations may explain why WhSJ maintained higher measurable protein content but a reduced pool of free amino acids [29,30].

From a technological perspective, this balance influences the textural properties of the jellies. Higher protein content can support a firmer gel network, contributing to structural stability, while the reduced free amino acid content may moderate flavor development compared with berry jellies richer in amino acid precursors. Nutritionally, the retention of intact protein fractions in WhSJ highlights the capacity of strawberry-derived compounds to protect whey proteins against degradation, though it may limit the immediate availability of free amino acids, which are more rapidly absorbed in the human body [31,32].

At the level of free amino acids (FAAs), the values were significantly higher in the jellies with added whey and fruits compared to the control, ranging from 230.78 ppm (CJ) to 1513.12 ppm (WhRJ). The literature reports that the addition of red fruits, rich in organic acids and polyphenols, can stimulate the availability of amino acids and positively influence protein digestibility [33]. Moreover, studies on protein gels or functional gels indicate that the addition of proteins such as whey promotes a significantly higher content of free amino acids in the final product [27].

Polyphenols, such as anthocyanins, flavonols, and phenolic acids abundant in berries, are known to interact with proteins through hydrogen bonding, hydrophobic interactions, and, in some cases, covalent linkages [34]. These interactions can lead to the formation of protein–polyphenol complexes, which influence solubility, digestibility, and the release of bioactive compounds. In the case of whey-based jellies, the extent of complex formation helps explain differences in protein and free amino acid contents across formulations. Stronger interactions may preserve protein integrity, while weaker interactions or enzymatic activity during processing can release small peptides and free amino acids [34].

From a digestive standpoint, the coexistence of intact whey proteins, peptides, and free amino acids in the same food matrix can be considered advantageous. While intact proteins ensure sustained amino acid release during digestion, free amino acids provide an immediate nutritional boost, balancing rapid absorption with prolonged availability. Therefore, differences in free amino acid concentrations among the whey–berry jellies are not only a marker of protein–polyphenol interactions but also have nutritional significance by influencing the kinetics of protein digestion and amino acid uptake [35].

Regarding total mineral substances, a significant increase was noted in jellies with added whey (WhJ, 0.34%) and strawberries (WhSJ, 0.35%), compared to the control (0.15%). This aspect correlates with the fact that whey contains significant amounts of calcium, magnesium, and phosphorus [26], and the fruits additionally contribute essential minerals and trace elements. Blueberry and raspberry jellies had intermediate values (0.26%), which reflects a variability dependent on the specific composition of each fruit.

The addition of fruit to the whey jelly causes significant changes in its chemical composition. Thus, comparing WhBJ vs. WhJ, it can be observed that the humidity remains practically the same, but proteins increase (+3.26%), minerals decrease markedly (−23.8%), while FAAs are comparable to WhJ. This results in protein fortification without a penalty on FAAs, but with lower mineralization of the jelly. Comparing WhSJ vs. WhJ, the most significant increase in proteins is observed (+4.88%), but FAAs decrease significantly (−9.59%), suggesting polyphenol–protein interactions (strawberries, ellagitannins/anthocyanins) that “bind” the protein fractions and reduce the measurable FAA pool. Minerals increase slightly (+5.06%).

Comparing WhRJ vs. WhJ, it can be observed that proteins increase moderately (+2.56%), while FAAs reach their peak (+6.33%), indicating either a higher extractability of amino acids in the raspberry matrix or less restrictive polyphenolic interactions, while mineral contents decrease (−23.8%).

The results confirm the data from the recent literature, which emphasizes that the use of whey as a functional ingredient and the enrichment with fruits rich in bioactive compounds represent an effective strategy for the development of food products with superior nutritional value and potential health benefits [28].

Figure 2 presents the evolution of the pH and titratable acidity of samples, initially and after storage for 7 and 14 days at 4 °C and ambient temperature (20–25 °C). It can be observed that, initially, the pH of the whey sample (WhJ) is relatively neutral to slightly acidic (5.6) and low titratable acidity (0.8) was recorded, because whey has a composition rich in proteins and lactose, but does not contain many free organic acids. Whey and fruit jellies (WhSJ/WhBJ/WhRJ) initially have a lower pH (3.86–4.40) and higher acidity (5.2–6.6), due to the citric, malic, and other natural organic acids found in berries. CJ is prepared only on the basis of water, sugar, and gelatin, so the initial pH is slightly neutral (6.5), and the acidity is very low, almost negligible, because there are no natural organic acids present as in fruits. The titratable acidity and pH of berry fruits were determined initially and the values obtained for pH were between 2.9 and 4.4, with the titratable acidity being between 1.3 and 3.5%, with blueberries being less acidic compared to raspberries and strawberries. The average values for pH in the case of sweet whey used in the production of jellies were 6.6, and the titratable acidity was 1.6%.

After 7 days of storage at 4 °C (refrigeration) the pH of the WhJ sample decreases slightly (by 0.3 units), while the acidity increases slightly (from 1.4 to 1.8). In the case of the WhSJ/WhBJ/WhRJ samples, the pH and acidity remain relatively stable, due to the inhibitory effect of polyphenols and anthocyanins on the microbial flora. The control sample (CJ) registers a slightly lower pH (6.35) due to possible minor degradation reactions of sugars and exposure to oxygen. The acidity remains very low.

In the case of storing the jellies under ambient conditions, after 7 days a more pronounced decrease in the pH of the WhJ sample (from 5.6 to 4.85) and a higher acidity (3.5) were observed, due to lactose fermentation. For the WhSJ/WhBJ/WhRJ samples, the pH decreases visibly and the acidity increases more rapidly, also favored by the degradation of polyphenols and organic acids in the fruit. For the control sample, CJ, the pH decreases by 0.4 units due to incipient Maillard reactions and possible microbial contamination. The acidity increases slightly, but the values remain low compared to fruit jellies.

After 14 days of storage at 4 °C, in the case of WhJ, the pH slowly decreases to 5.0 and the acidity increases to 2.5, while for the WhSJ/WhBJ/WhRJ samples the pH remains in the range of 3.2–3.8 and the acidity stabilizes, being limited by the synergistic effect between the bioactive compounds and the cold storage conditions. For the control sample, the pH is 5.8, very low acidity, and the product remains stable due to the high sugar content (preservative effect).

At ambient conditions, the pH for WhJ reaches 4 units, significantly increased acidity, while for WhSJ/WhBJ/WhRJ samples, pH drops below 3.5. The control sample registers a pH of 5.5 and the acidity increases slightly.

Previous studies on jams and jellies indicate a slow decrease in pH and a progressive increase in acidity during storage at room temperature; at refrigeration, the changes are much slower [36,37]. Brandão et al. observed pH values in the range of 4.0–4.5 for sugar-free jams that lost only a few tenths in 60 days, with minimal variations in acidity (0.42–1.12%). The closer the initial pH is to the neutral zone, the slower the pH changes during storage; fermentable compounds are absent, thus microbial activity is limited [38].

### 3.2. Macro- and Microelement Contents of Whey Jellies

In Table 3 the macro- and microelement compositions of jellies fortified with whey and fruits are presented.

The macroelement composition of whey showed high contents of K (3877.90 ppm), Ca (1477.44 ppm), and Mg (589.54 ppm). The berry juice has a lower contribution to the total macroelement composition and varies between 121.52 and 843.20 ppm. Regarding microelements, whey recorded the highest values for Zn (10.32 ppm) and Fe (7.91 ppm), strawberry for Mn (2.76 ppm) and Cu (1.77 ppm), blueberry for Cr (1.98), and raspberry for Ni (1.43 ppm).

Whey-added jellies (WhJ) showed the highest values for potassium (4184.33 ppm), calcium (1693.33 ppm), magnesium (696.51 ppm), and sodium (23.34 ppm). This confirms the rich mineral profile of whey, recognized as an important source of easily bioavailable soluble minerals, especially calcium, magnesium, and potassium [28].

Fruit jellies showed intermediate values, especially WhSJ (2179 ppm), confirming the additional contribution of fruits to potassium intake.

In terms of calcium (Ca), whey brought a remarkable contribution (WhJ—1693 ppm) compared to the control (129 ppm). The high calcium content is specific to sweet whey, which contains 70–80% of milk minerals, especially calcium and magnesium [39].

Fruit jellies had values between 762 and 833 ppm, indicating that the addition of berries contributes partially to the mineral intake, but that whey remains the main source.

Magnesium (Mg) followed a similar pattern: it was highest in WhJ (697 ppm), then in WhSJ (596 ppm), WhBJ (472 ppm), and WhRJ (411 ppm), compared to CJ (182 ppm). Magnesium is present in both whey and berries, and studies show that fruits such as strawberries and blueberries are important natural sources of magnesium and potassium [40].

Sodium (Na) was very high in all jellies with added whey and fruit (23–49 ppm), and absent in CJ, reflecting the contribution of whey and fruit to natural sodium intake.

Regarding the microelements, whey jellies with fruits showed clearly higher concentrations compared to the control (CJ), where most were not detected. Zinc (Zn) and iron (Fe), essential minerals for the immune system and enzymatic processes, had the highest concentrations in WhJ: 16.36 ppm Zn and 16.16 ppm Fe, followed by WhBJ (Zn: 4.66 ppm, Fe: 11.21 ppm).

Manganese (Mn) and copper (Cu) were higher in fruit jellies, WhSJ—2.23 ppm Mn, 1.57 ppm Cu; WhBJ—1.60 ppm Mn, 0.87 ppm Cu, compared to WhJ, suggesting that these trace elements come predominantly from berries.

Trace elements such as Ni and Cr were detected at low levels (0.7–1.2 ppm) only in fruit jellies, confirming the literature showing that berries can provide rare trace elements such as Ni and Cr [41]. The control jelly (CJ) showed intermediate Ni content, suggesting a minor background from non-berry ingredients or matrix effects, whereas adding whey alone did not increase Ni content (WhJ). Cr was not detected in CJ and WhJ but was consistently detected—and at comparable magnitudes—in all berry-containing jellies, indicating that berries are the principal source of both Ni and Cr in these formulations. This aligns with known trace metal uptake by small fruits from soil and irrigation water; in contrast, the absence of Cr in WhJ argues against appreciable process-equipment leaching under the applied conditions.

From a safety/quality standpoint, the whey does not elevate Ni/Cr, while the berry component dictates trace metal presence; therefore, raw material sourcing and agronomic provenance (soil, cultivar, and farming practice) are the key levers to manage Ni/Cr levels [41].

As an overview we can say that WhJ presented the highest levels of K, Ca, Mg, Zn, and Fe, confirming the role of whey as a major source of minerals. WhSJ, WhBJ, and WhRJ completed the mineral intake, especially through microelements (Mn, Cu, Ni, and Cr), which reflects the mineral composition specific to berries.

### 3.3. Phytochemical Profile of Whey Jellies

In Table 4 the phytochemical profiles (TPC and AA) of jellies fortified with whey and fruits are presented.

Whey is a matrix with a low content of total polyphenols, with the contribution of phytochemicals being provided by berries, where the TPC content varied between 139.44 mg/100 g in raspberries and 193.54 mg/100 g in strawberries.

The results show that the control jellies (CJ) did not present detectable polyphenols, while those with whey (WhJ) had a very low content (6.79 mg/100 g). This confirms that whey has a low content of phenolic compounds, instead being a source of proteins and minerals. Whey can contain phenolic compounds, predominantly benzoic acid derivatives (e.g., p-hydroxybenzoic, vanillic, syringic, and salicylic) and hydroxycinnamic acids (e.g., p-coumaric, ferulic, caffeic, and sinapic), but native levels are low and depend on an animal’s diet, and they increase or diversify when whey is fermented or combined with plant ingredients. In a goat study that analyzed milk, whey, and cheese by HPLC-DAD, whey showed a profile rich in benzoic acid derivatives, while plant flavones present in feed were not found in milk/whey—evidence that simple phenolic acids are the dominant transferrable species [42]. In contrast, the addition of berries significantly increased the level of polyphenols: WhSJ (196.48 mg/100 g) has the highest content, followed by WhBJ (145.32 mg/100 g) and WhRJ (136.7 mg/100 g). These values confirm that berries are major sources of polyphenols (anthocyanins, flavonols, ellagic acid, etc.), bioactive compounds recognized for their antioxidant and anti-inflammatory properties [42]. In this regard, Hwang, H. et al., 2020 mentioned that fresh blueberries have a total content of polyphenols between 196 and 255 mg/100 g FW [43].

The antioxidant activity followed a similar pattern to that of polyphenols. The values were very low in WhJ (2.45%) and WhRJ (3.20%), confirming that the simple addition of whey or raspberries does not provide antioxidant activity comparable to that of other fruits. While most raspberry anthocyanins are antioxidants, the glycosylation pattern can reduce activity substantially (e.g., certain 3-glycosides/arabinosides show markedly lower activity vs. aglycones in DPPH/ABTS/FRAP), meaning little or undetectable effect on typical test concentrations depending on the assay/design. Also, native raspberry ellagitannins are potent antioxidants in vitro, but their bioavailable forms (urolithins, esp. Uro-B and methylated urolithins) often lack direct radical scavenging in standard chemical assays [44,45]. The highest levels were observed in WhBJ (28.11%) and WhSJ (25.86%), in direct correlation with the high polyphenol content. The literature shows that anthocyanins in blueberries and strawberries are responsible for the strong antioxidant activity, contributing to the neutralization of free radicals and protection against oxidative stress [41]. In contrast, raspberry-specific polyphenols have lower activity in this type of food matrix, possibly due to interactions with whey proteins that may limit their bioavailability [42].

For blueberries, the reported average polyphenol content is 196–255 mg/100 g FW, and DPPH efficiency reaches radical inhibition values between 90 and 93% [43]. In wild blueberry samples, TPC varies between 424 and 819 mg GAE/100 g fresh weight (FW) and correlates very strongly with DPPH activity (r ≈ 0.92–0.99) [46].

Thus, the experimental results align with the documented trend: WhSJ and WhBJ reach the highest levels of polyphenol and antioxidant activity, while WhRJ has somewhat more moderate values, but still superior to those of the control.

### 3.4. Storage Stability of Whey Jellies

The storage stability of jellies was determined by the syneresis test, the water loss (%) during storage at different temperatures being presented in Figure 3. This method allowed for the assessment of the water behavior of gels over time, providing valuable information about the structural resistance of the gel and the possible influence of functional ingredients (such as whey or fruit juice) on syneresis [47]. The samples were kept at constant temperatures of 4 °C and 20–25 °C and analyzed at intervals of 7 and 14 days.

The control sample (CJ) initially presents a stable gel network, without additional proteins or polyphenols with stabilizing roles. After 7 days at 4 °C, the syneresis process is minimal (3%) and the gel remains compact, while at 20–25 °C after 14 days it exceeds 5%, indicating destabilization by gel contraction. The WhJ sample records at 4 °C very low water loss (2.1%), and even after 14 days very good stability is observed, with a low syneresis process (4.1%). Whey proteins interact with the polysaccharides in the gel (gelatin), increasing the water retention capacity [27]. At 20 °C and 14 days the water loss increases (5.2%), but is still acceptable.

Lower syneresis at 4 °C is consistent with a tighter, colder gel network that restricts water mobility; in contrast, storage at 20–25 °C increases molecular mobility and promotes serum release. In mixed protein–hydrocolloid gels, water holding capacity (WHC) and syneresis are inversely related and track a gel’s microstructure (pore size/rigidity) and bound vs. free water fractions. Studies on dairy/acidified milk model systems show that added whey proteins decrease water mobility and improve WHC, and that microstructural indices (e.g., T_2_ relaxation) correlate with lower syneresis during storage [26].

In gelatin-based gels, incorporating globular proteins (including whey proteins) enhances water binding and processing yield, yielding softer but more water-retentive networks; WHC improvements are repeatedly reported for mixed gelatin–soy/whey systems and for gelatin gels fortified with functional proteins or mild cross-linking [27].

The fruits in the samples WhSJ, WhBJ, and WhRJ add organic acids and polyphenols that can influence the gel structure. WhSJ and WhBJ have a high content of polyphenols and can strengthen the protein–polysaccharide interactions, which confers good stability and a loss of <5% at 4 °C, even after 14 days of storage, and of around 5% at 20–25 °C after 14 days. The sample WhRJ has a higher content of organic acids, which can destabilize the polymer network. At 4 °C good stability is recorded (loss of 4.8% after 14 days), while at 20–25 °C the loss is higher than 5%, showing a tendency towards instability.

The syneresis test confirms that WhJ and WhBJ are the most stable samples due to protein–polyphenol interactions, which reduce syneresis. WhSJ showed moderate stability with a resistant network, but was slightly sensitive to storage at 20–25 °C. The least stable jelly was WhRJ, where a higher content of organic acids can cause water migration. CJ is stable at 4 °C, but at 20–25 °C after 14 days can exceed the 5% threshold, showing lower stability than the protein/fruit samples.

### 3.5. Color Analysis of Whey Jellies

Color is one of the first factors that influence a consumer’s perception of the quality of a food product, including jellies. In the case of jellies, color reflects not only the composition of the ingredients (such as natural extracts, fruit juices, dyes, or caramelized sugar), but also possible changes that have occurred during processing or storage [48].

Color determination represents an important tool in quality control and product standardization, as color modifications may reflect the degradation of thermolabile compounds such as anthocyanins and carotenoids, as well as changes due to Maillard reactions or oxidative processes [49]. Consequently, the evaluation of color parameters in jelly products provides essential information regarding both consumer acceptability and the chemical stability of bioactive ingredients throughout processing and storage.

Table 5 presents the color parameters of jellies samples analyzed using the CIELAB system. The color profile was influenced by the formulation: products containing red fruit juices (e.g., strawberry, raspberry) typically exhibited high positive *a** values, whereas those formulated with blueberry extracts showed negative *b** values, associated with bluish-purple tones.

The measured CIELAB values perfectly reflect the impact of fruit anthocyanins and protein–pigment interactions on the color of jellies.

The highest brightness (*L**) value was recorded for the control sample, CJ (46.14), which has a light appearance, specific to jellies without added whey or fruit. The addition of fruits rich in anthocyanins reduces brightness, generating more intense coloration. The lowest value (18.46) was recorded for the sample WhBJ, a dark jelly, explained by the anthocyanin pigments in blueberries. Jellies with whey and fruit (WhSJ, WhRJ) have intermediate values (≈30–32), indicating darker shades than the control, but lighter than WhBJ. Brightness (*L**) significantly decreases as the anthocyanin content increases—lower values indicate a more intense, saturated color. This pattern is confirmed in the literature: matrices containing pigmented anthocyanins in red-orange or violet colors become darker during storage due to pigment degradation and negative copigmentation that reduces light reflectance [50].

The lowest value for the red-green component (*a**) was recorded for CJ (–1.70), which has a very weak greenish tint. Very high values were observed for WhSJ (23.22) and WhRJ (23.17) due to the intense red shades characteristic of strawberries and raspberries and confirm the presence of red anthocyanins. WhJ (19.5) was also found to be in the red zone, but less intense, while WhBJ’s (5.88) color is weaker red, tending towards violet-blue.

High positive values for *a** are directly correlated with the anthocyanin content in raspberry and strawberry jellies. In food systems that contain anthocyanins (e.g., beverages or fermented dairy products), the degradation of colorants is often associated with a decrease in value over time, which reflects the loss of red intensity [51,52].

Regarding the yellow-blue component (*b**), all samples have positive values, a yellow tint, but the intensity differs. The highest value was recorded for WhSJ (8.88), with a red-yellow (orange-reddish) tint. The value for WhRJ was 5.5, with a weak yellow color, while the color of the WhBJ sample (7.67) was balanced between red and blue, justifying the purple tint of blueberries. The positive parameter *b** is typical for red fruits with anthocyanins; however, the exact composition influences the degree of yellow. In products based on blueberries, *b** tends toward lower values due to the dominant blue pigments (delphinidin) and copigmentation with proteins [53].

Chroma (*C**) reflects saturation. Anthocyanin-rich systems may exhibit high *C** values, but these decrease over time due to pigment degradation, especially in the presence of oxygen, higher pH, or heat [54]. WhSJ (24.86) and WhRJ (23.81) have the most saturated colors, intense and stable red, while WhBJ (9.66) shows less saturated color and duller tones, associated with violet-blue, and CJ (4.84) has a very reduced color, almost colorless.

Hue angle (h°) provides information about the overall hue: low values (<20°) indicate intense red hues, and higher values (>50°) approach violet-blue [54]. In food systems with blueberry anthocyanins, *h*° gradually increases during storage, reflecting the migration of hue towards blue-green as the anthocyanins degrade [54]. In our study, *h°* values confirm that raspberry and strawberry jellies fall into the deep red zone, while blueberries tend towards purple. The results confirm the yellow-greenish tint (actually almost neutral) of the CJ sample (110.57), deep red, a slightly purplish one for WhRJ due to the lower value (13.35), and red for WhSJ (20.93) and WhJ (18.86). The value recorded for WhBJ (52.53) indicates a color between red and violet-blue.

### 3.6. Texture Analysis of Whey Jellies

The texture parameters are presented in Table 6. Texture determination is essential not only for consumer acceptability, but also for product stability during storage and handling.

The analysis of textural parameters reveals significant differences between samples, suggesting an important influence of composition on the consistency and sensory quality of the products.

Hardness shows increasing values from CJ (0.22 ± 0.01 N) to WhBJ (0.32 ± 0.03 N), which suggests that the addition of protein and polyphenolic ingredients leads to a more compact and resistant gelled network. This result is consistent with the observations of Ersch et al. (2016), who showed that interactions between proteins and gelatine increase the rigidity and resistance of food gels [55]. Duncan’s test was performed to compare group means. Significant differences (*p* < 0.05) were found between the control and all other jelly groups, but no significant differences were found between fruit jellies and whey jelly.

Adhesiveness is very low in CJ (0.13 ± 0.06 mJ), but increases significantly in the case of samples with the addition of proteins and fruits rich in bioactive compounds (WhBJ: 0.66 ± 0.21 mJ). This increase is attributed to interactions between polyphenols and proteins, which enhance intermolecular bonds and influence a gel’s ability to adhere to surfaces, a phenomenon also described by Yan et al. (2025) in their study on protein–polyphenol synergy [56].

Cohesiveness ranges from 0.29 (WhJ) to 0.49 (WhRJ), with higher values in samples containing red fruits (WhRJ). Higher cohesion indicates a uniform internal structure, characteristic of products with consolidated protein networks, an aspect supported by the results of Wang et al. (2022), who reported that polyphenols stabilize protein structures and influence rheological properties [57].

Elasticity (springiness index) is maintained at moderate levels (0.53–0.93), with the highest value being in WhBJ (0.93 ± 0.03). This suggests that the addition of blueberries or fruits rich in anthocyanins may contribute to the elasticity of the gel, similar to the observations of Chima et al. (2022), who showed that blueberry polyphenols significantly influence the mechanical properties of dairy and vegetable products [35]. Significant differences were observed between the control and all fruit jellies (*p* < 0.05), but not between the control and whey jellies and between fruit jellies.

Gumminess and chewiness show notable variations: CJ has the lowest values (0.09 N and 0.33 mJ), and WhBJ the highest (0.15 N and 1.7 mJ). These differences indicate that samples enriched with polyphenols and proteins have a denser texture and are more resistant to chewing, consistent with the results of Zhou and Pang (2018), who emphasized the role of electrostatic interactions in the stability and hardness of protein structures [58].

When the primary network (gelatin + whey proteins) and the total solids are controlled, the type of berry juice acts as a fine modulator of cohesiveness, elasticity, and adhesion through protein–polyphenol interactions and environmental effects (pH/ions/pectin/water), while the hardness remains dominated by the gel base and therefore does not significantly differ between fruits (Cassas, 2020) [59].

Berry fruit juices bring pH, ions (K⁺, Ca^2^⁺), and small amounts of pectin that alter the charge and electrostatic screening in the network, affecting elastic recovery (springiness) and cohesion, without necessarily altering maximum compressive strength. The literature shows that moderate variations in polysaccharide/ionic composition can shift brittleness and adhesiveness, leaving ‘hardness’ almost constant when the gel base (gelatin/protein) is identical (Spahn, 2008, Said 2023) [60,61].

Overall, the data show that samples enriched with anthocyanin- and polyphenol-rich fruits (WhBJ, WhRJ) have a firmer, more elastic, and resilient texture, which gives them superior textural characteristics compared to the control (CJ). This effect is associated with the complex interactions between proteins and polyphenols, which contribute to the formation of a stable three-dimensional network.

### 3.7. FTIR Spectra Analysis of Whey Jellies

Figure 4 presents the FTIR spectra of whey jellies. The spectra confirm that each sample exhibits the characteristic bands corresponding to the ingredients used.

The interpretation of FTIR spectra for whey jelly samples with strawberries, blueberries, and raspberries highlights the presence of specific functional compounds from the natural ingredients and the protein base (whey), which can influence the physicochemical and functional properties of the jellies. The analysis is based on comparing the characteristic bands for each compound and identifying significant variations between the samples.

The FTIR spectra of the analyzed jelly confirm the presence of whey proteins, carbohydrates, and bioactive compounds from fruits. The region of 3200–3600 cm^−1^ is dominated by a broad and intense band characteristic of O–H and N–H stretching vibrations, indicating the presence of hydroxyl groups (from polysaccharides, water, and polyphenols) and amines (from whey proteins). The relatively high intensity of this band in fruit samples (especially strawberries and raspberries) can be attributed to the high content of phenolic compounds and bound water, as reported in the literature [62].

The region of 2800–3000 cm^−1^ corresponds to C–H (aliphatic) bonds. The bands observed around 2924 and 2854 cm^−1^ correspond to the symmetric and asymmetric stretching vibrations of methyl and methylene chains. These are typical for lipid components or carbohydrate structures, but can also originate from protein structures in jellies [62].

The intense band around 1634–1645 cm^−1^ is associated with the stretching vibrations of the C = O bonds in amide groups (Amide I), characteristic of whey proteins. Slight variations in position or intensity between samples may indicate protein–polyphenol interactions or changes in the secondary structure of the proteins [62].

The presence of bands around the value of 1530–1545 cm^−1^ is specific to the stretching vibrations of N–H and C–N (Amide II), confirming the presence of proteins. The intensity of the bands is higher in the ‘Whey’ sample, confirming its predominant protein nature [63].

The region of 1000–1200 cm^−1^ is characterized by multiple bands, especially around 1040–1075 cm^−1^, indicating the presence of carbohydrates (pectins, sugars) and phenolic compounds in fruits. The presence of these bands is more pronounced in samples with strawberries, blueberries, and raspberries, suggesting a significant contribution of the plant matrix [63].

The appearance of bands in the region of 600–900 cm^−1^ can be attributed to C–H vibrations and other specific modes of aromatic compounds in fruits. The differences between samples in this region reflect the different phytochemical composition of the fruits (anthocyanins, flavonoids, etc.) [64].

### 3.8. Microscopic Analysis of Whey Jellies

Figure 5 presents images of jellies using simple optical microscopy.

The microscopic analysis of gels highlighted significant structural differences depending on their composition, particularly due to the presence of whey and fruits. The images obtained through simple optical microscopy (Figure 5) allowed for the observation of the homogeneity of the gelatinous matrix, the dispersion of the added components, and the structural porosity. In the case of the control sample (Control), which contained only gelatine, the structure appeared to be uniform, compact, and relatively translucent, characteristic of a homogeneous matrix formed exclusively from a collagen network (Figure 5A. This organization is typical of pure protein gels, with a well-defined network and devoid of inclusions or agglomerates [56].

The WhJ jelly exhibited a more porous structure compared to the control, with visible distributions of whey protein particles. These were unevenly dispersed, suggesting slight interference in the formation of the gelatin network, likely due to competition between whey proteins and collagen for hydrogen bonds and electrostatic interactions during gelation [57].

In the case of the WhBJ jelly (with whey and blueberries), the structure highlighted local aggregations and visible inclusions of anthocyanin pigments, indicating a partially homogeneous distribution of the blueberry extract. This organization suggests a partial interaction between the polyphenols from the blueberries and the gel proteins, with the potential formation of protein–polyphenol complexes, a fact also supported by the specialized literature [35].

The WhRJ jelly (with whey and raspberry) exhibited a visibly denser structure, with dark-colored areas, possibly due to the higher concentration of phenolic compounds and pectin from the raspberries, which may contribute to strengthening the gelatinous network. This densification can be explained by the synergy between the soluble polysaccharides from raspberries and gelatin, which contribute to a more compact three-dimensional network [35].

The WhSJ jelly (with whey and strawberries) showed an intermediate structure between the other fruit formulations, with moderately distributed inclusions and a relatively porous network. The presence of strawberries induced a more airy texture, possibly due to the lower content of insoluble fibers and the different profiles of phenolic compounds, which influences the degree of gelatin cross-linking [58].

These observations suggest that the addition of whey and fruits significantly affects the microtextural structure of the jellies, which may influence their sensory and functional characteristics.

### 3.9. Correlation Between Parameters

#### 3.9.1. Pearson Correlation

In Figure 6 is presented the Pearson correlation between different physical–chemical parameters of jellies with whey and fruits.

The correlation analysis (Figure 6) indicates strong and positive correlations (r > 0.8), with a direct association between polyphenols and Fe (0.85), Mn (0.82), Ni (0.85), proteins (0.81), and amino acids (0.83). This suggests that the addition of polyphenol-rich ingredients (e.g., fruit juice) simultaneously contributes to increasing the mineral and protein contents of the gels [65]. The high positive correlation between polyphenols and DPPH antioxidant activity (r = 0.78) indicates that fruit juice is the main source of bioactive compounds with an antioxidant role [66]. This association is supported by FTIR data, which revealed specific bands of phenolic –OH groups (~3300 cm^−1^) and C = O (~1720 cm^−1^). Proteins and amino acids show extremely high correlations with the textural parameters: cohesiveness (0.96), springiness (0.91), gumminess (0.97), and chewiness (0.96), which suggests that a well-developed protein matrix favors the elastic, gummy texture and optimal chewability of the gel. Microscopic observations confirmed a dense and uniform gel network in the samples with added whey, which explains the high values of hardness and elasticity [67]. FTIR revealed changes in the Amide I (~1650 cm^−1^) and Amide II (~1540 cm^−1^) bands, indicating interactions between proteins and polyphenols.

Texture parameters are closely related to each other, indicating that changes in one mechanical property (e.g., hardness) are also reflected in the others: cohesiveness–chewiness (0.97), gumminess–chewiness (0.98).

Moderate correlations (0.5–0.8) suggesting a positive but not absolute relationship were recorded between DPPH and polyphenols (0.78), which suggests that antioxidant activity (measured by DPPH) is significantly correlated with polyphenol content, but not perfectly, possibly due to other antioxidant compounds (e.g., vitamins, amino acids).

Minerals among themselves (Ca, Mg, Zn, Fe, Mn, Cu, and Na) also record moderate correlations, indicating that the added source (whey, fruit juice) simultaneously provides several mineral elements.

Notable negative correlations indicating an inverse relationship—when one variable increases, the other decreases—were recorded between the pairs polyphenols–Cr (−0.65) and adhesiveness–proteins/amino acids (−0.75, −0.65), which suggests that increasing protein content reduces gel adhesiveness, which may improve handling and sensory acceptability. This phenomenon is supported by microscopic images, where protein-rich samples showed a more compact network with smaller pores, which limits exudation and surface adhesion.

#### 3.9.2. Principal Component Analysis and Cluster Analysis

The inertia of the first principal components shows if there are strong relationships between variables and suggests the number of components that should be studied. The first two principal components of analysis express 83.48% of the total dataset inertia; this means that 83.48% of the individual’s cloud total variability is explained by the plane spanned by the first two principal components (Figure 7A). This percentage is very high; thus, the first plane represents data variability very well. This value is greater than the reference value; the variability explained by this plane is thus highly significant (the reference value is the 0.95 quantile of the inertia percentage distribution obtained by simulating data tables of equivalent size based on a normal distribution). An estimation of the right number of axes to interpret suggests restricting the analysis to the description of the first two axes. These axes present an amount of inertia greater than those obtained by the 0.95 quantile of random distributions (83.48% against 73.25%). This observation suggests that only these axes carry real informations [68].

The first principal component factor is significant: it expresses itself in 50.82% of the data variability (Figure 7B). This observation suggests that this axis carries great information. The most important contribution for this component comes from the following variables: springiness, Cr, proteins, Na, hardness, polyphenols, chewiness, gumminess, adhesiveness, Ni, Mn, and DPPH are highly correlated with this dimension These variables could therefore summarize themselves as dimension 1 (Figure 7C).

The second principal component expresses itself in 32.66% of the data variability (Figure 7B). The most important contribution of this component comes from the variables Ca, Zn, K, Mg, Cu, Fe, and cohesiveness (which are all highly correlated) (Figure 7D).

The hierarchical classification, represented by a dendrogram (Figure 8), grouped jelly samples into three distinct clusters. The first cluster, consisting of WhRJ, WhSJ, and WhBJ samples, indicates a high similarity among them, which can be attributed to similar compositional characteristics. The second cluster includes the WhJ sample, which presents an intermediate profile, different from the previous group but not entirely divergent. The third cluster, represented solely by the CJ sample, is the farthest from the others, suggesting a distinct profile, likely determined by variations in formulation, ingredients, or physicochemical properties.

By combining the results of PCA and cluster analysis, it is confirmed that jelly samples can be objectively differentiated based on the measured characteristics, which facilitates the identification of similarities and differences between types and can guide the development or optimization of products.

## 4. Conclusions

The incorporation of whey and berries into jellies demonstrates an innovative approach to developing functional foods with enhanced nutritional quality, enriched in macro- and microelements as well as bioactive phytocompounds essential for human health. Beyond improving textural stability—where protein–polyphenol interactions in WhJ and WhBJ reduced syneresis and ensured better network resistance compared to WhSJ—the study highlights the broader potential of valorizing whey, a by-product of the dairy industry often regarded as waste. By transforming it into a valuable ingredient, this research advances sustainable food innovation, reduces environmental burden, and supports the transition toward a circular economy. Storage trials further confirmed that refrigeration at 4 °C ensures higher stability, while ambient conditions accelerate decreasing pH and increasing acidity. Overall, the synergistic use of whey and berries not only enhances microstructure, but also exemplifies a sustainable strategy for mitigating food waste, aligning with global and European objectives for health promotion, resource efficiency, and sustainable development.

## Figures and Tables

**Figure 1 foods-14-03193-f001:**
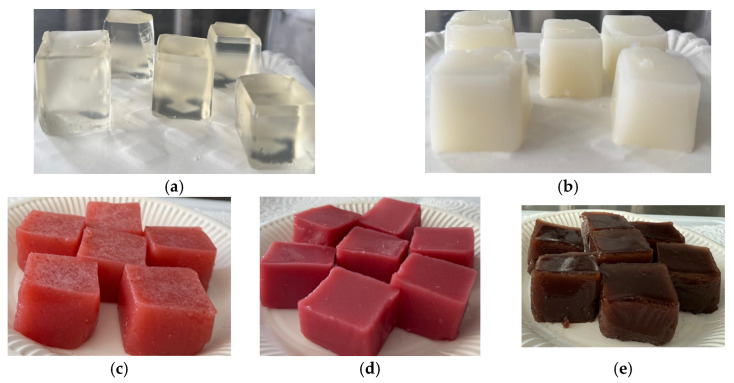
The jellies: (**a**) CJ; (**b**) WhJ; (**c**) WhSJ; (**d**) WhRJ; and (**e**) WhBJ.

**Figure 2 foods-14-03193-f002:**
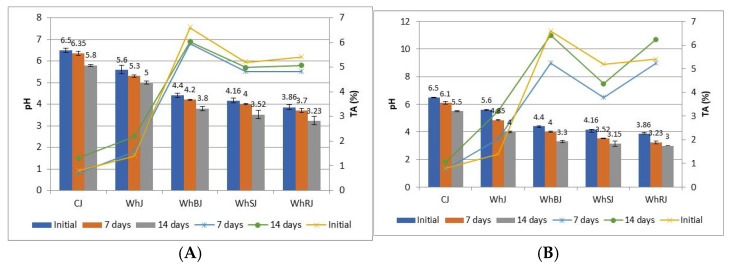
Evolution in time (0–14 days) of pH and titratable acidity (TA) of jelly samples. (**A**) Jelly samples stored at 4 °C. (**B**) Jelly samples stored at ambient temperature (20–25 °C). The columns express pH values, and the lines expresses titratable acidity (TA) values.

**Figure 3 foods-14-03193-f003:**
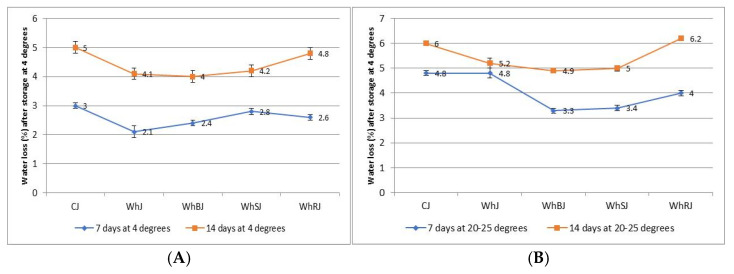
Syneresis curves of jelly samples in terms of time (0−14 days) expressed as water loss (%) after storage at 4 °C (**A**) and at ambient temperature (20−25 °C) (**B**).

**Figure 4 foods-14-03193-f004:**
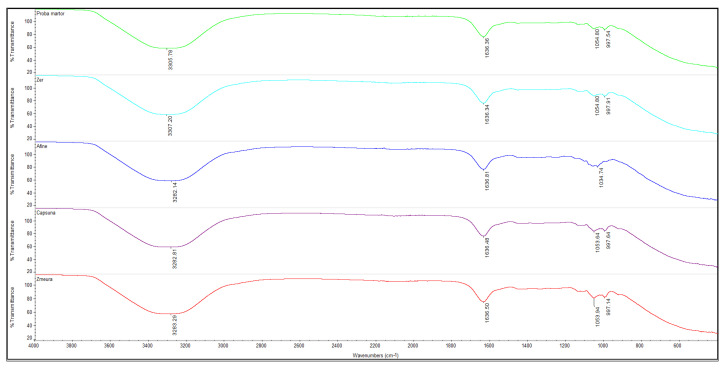
FTIR spectra of jellies: CJ (green line), WhJ (light blue line), WhBJ (dark blue line), WhSJ (purple line), and WhRJ (red line). Spectral range of 4000–400 cm^−1^; 32 scans at a 4 cm^−1^ resolution.

**Figure 5 foods-14-03193-f005:**
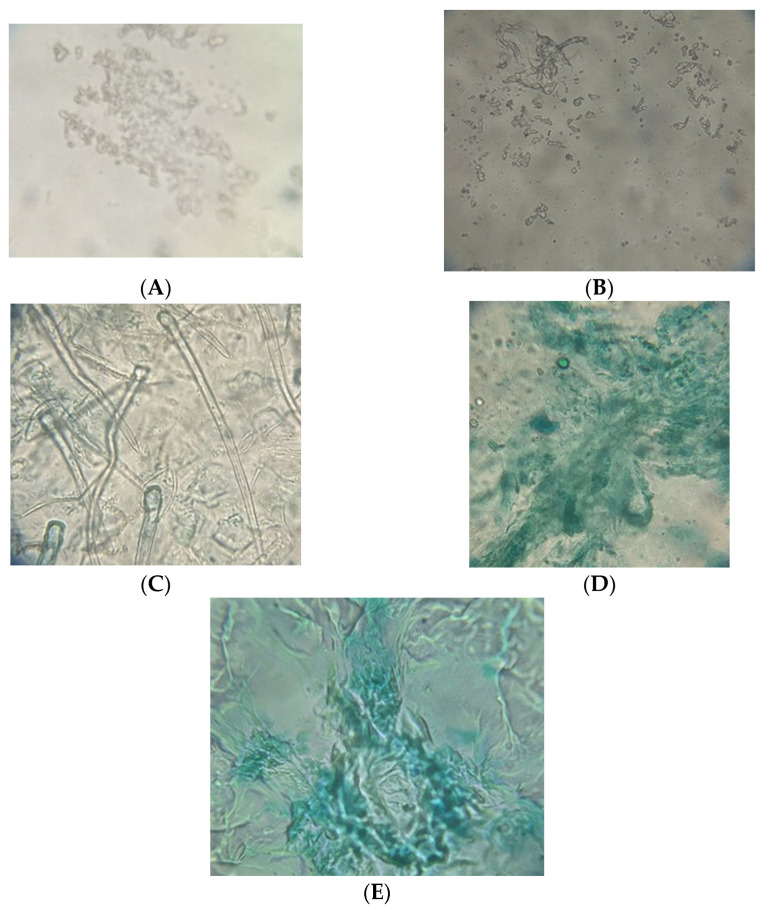
Microscopic analysis of jelly fortified with whey and fruits using 400× objective: (**A**)—Control, (**B**)—WhJ, (**C**)—WhRJ, (**D**)—WhBJ, (**E**)—WhSJ.

**Figure 6 foods-14-03193-f006:**
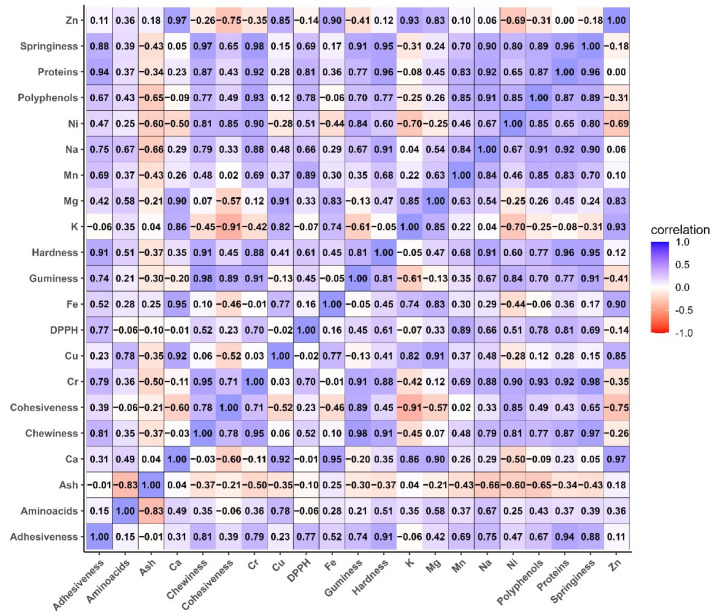
Pearson correlation between physical–chemical, macro- and microelement, and phytochemical parameters of jellies fortified with whey.

**Figure 7 foods-14-03193-f007:**
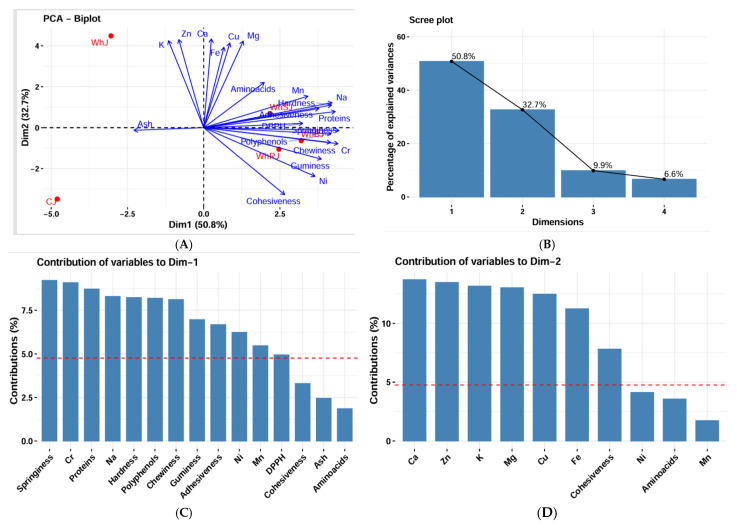
(**A**) Scree plot of PCA. (**B**) Biplot of PCA. (**C**) Contribution of variables to the first dimension of PCA. (**D**) Contribution of variables to the second dimension of PCA.

**Figure 8 foods-14-03193-f008:**
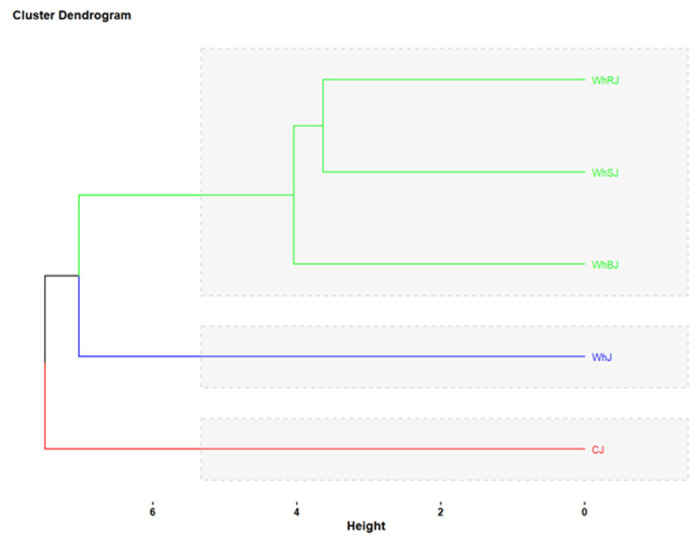
Cluster dendrogram of jellies fortified with whey.

**Table 1 foods-14-03193-t001:** Recipes for jellies.

Sample	Abbreviation	Berry Juice (mL)	Whey (mL)	Gelatin (g)	Sugar (g)	Water (mL)
Control	CJ	-	-	10	80	550
Whey jelly	WhJ	-	200	10	80	350
Whey strawberry jelly	WhSJ	300	200	10	80	50
Whey raspberry jelly	WhRJ	300	200	10	80	50
Whey blueberry jelly	WhBJ	300	200	10	80	50

**Table 2 foods-14-03193-t002:** Moisture, proteins, free amino acids, and total mineral substances of raw materials and whey jellies.

Samples	Moisture (%)	Protein (%)	Free Amino Acids (ppm)	Total Mineral Substances (%)
		Raw Materials		
Whey	93.40 ± 0.10 ^A^	1.10 ± 0.15 ^C^	800.54 ± 4.50	0.120 ± 0.25 ^C^
BJ	84.80 ± 2.34 ^C^	1.60 ± 0.04 ^B^	746.90 ± 2.50	0.134 ± 0.05 ^B^
SJ	90.35 ± 1.50 ^B^	1.72 ± 0.01 ^A^	545 ± 1.05	0.205 ± 0.02 ^A^
RJ	85.50 ± 2.50 ^C^	1.56 ± 0.02 ^B^	770 ± 3.05	0.130 ± 0.05 ^B^
		Jellies		
CJ	70.48 ± 0.34 ^a^	4.18 ± 0.05 ^e^	230.78 ± 2.00 ^e^	0.346 ± 0.05 ^a,b^
WhJ	62.34 ± 0.23 ^b^	4.30 ± 0.02 ^d^	1423.01 ± 2.51 ^b^	0.336 ± 0.01 ^b^
WhBJ	63.22 ± 0.25 ^b^	4.44 ± 0.05 ^b^	1447.04 ± 1.64 ^b^	0.256 ± 0.02 ^c^
WhSJ	62.86 ± 0.96 ^b^	4.51 ± 0.10 ^a^	1286.48 ±2.40 ^d^	0.353 ± 0.03 ^a^
WhRJ	62.70 ± 0.83 ^b^	4.41 ± 0.05 ^c^	1513.12 ± 0.95 ^a^	0.256 ± 0.01 ^c^

The values are expressed as the mean ± standard deviations. Within the data, different letters in the same column indicate significant difference (*p* < 0.05), according to Duncan’s test.

**Table 3 foods-14-03193-t003:** The macro- and microelement composition of raw materials and jellies fortified with whey and fruits.

Samples	Macro- and Microelements (ppm)									
	K	Ca	Mg	Zn	Fe	Mn	Cu	Na	Ni	Cr
Raw Materials										
Whey	3877.90 ± 1.76 ^A^	1477.44 ± 2.44 ^A^	589.54 ± 1.00 ^A^	10.32 ± 0.14 ^A^	15.14 ± 0.20 ^A^	1.24 ± 0.10 ^B^	0.90 ± 0.10 ^B^	20.33 ± 1.22 ^D^	nd	nd
SJ	843.20 ± 1.44 ^B^	121.52 ± 3.22 ^D^	518.68 ± 1.45 ^B^	nd	nd	2.76 ± 0.20 ^A^	1.79 ± 0.04 ^A^	66.67 ± 2.34 ^A^	1.24 ± 0.04 ^B^	1.63 ± 0.01 ^C^
BJ	450.22 ± 4.40 ^C^	259.22 ± 2.44 ^B^	322.45 ± 0.20 ^C^	nd	7.91 ± 0.10 ^B^	1.21 ± 0.40 ^B^	0.62 ± 0.02 ^C^	53.29 ± 2.90 ^B^	1.18 ± 0.02 ^C^	1.98 ± 0.04 ^A^
RJ	320.10 ± 2.23 ^D^	156 ± 1.33 ^C^	220 ± 1.22 ^D^	nd	nd	0.42 ± 0.01 ^C^	1.77 ± 0.04 ^A^	42.44 ± 0.55 ^C^	1.43 ± 0.01 ^A^	1.83 ± 0.04 ^B^
Jellies										
CJ	372.67 ± 7.76 ^e^	128.67 ± 14.57 ^d^	181.66 ± 1.52 ^e^	nd	nd	nd	nd	nd	0.39 ± 0.10 ^b^	nd
WhJ	4184.33 ± 0.58 ^a^	1693.33 ± 0.58 ^a^	696.51 ± 0.50 ^a^	16.36 ± 0.1 ^a^	16.16 ± 0.01 ^a^	1.44 ± 0.01 ^b^	1.24 ± 0.09 ^a^	23.34 ± 0.20 ^d^	0.01 ± 0.01 ^c^	nd
WhSJ	2179.54 ± 0.71 ^b^	762.46 ± 0.75 ^c^	595.72 ± 0.41 ^b^	4.39 ± 0.46 ^b^	5.62 ± 0.44 ^c^	2.23 ± 0.31 ^a^	1.57 ± 0.25 ^a^	49.34 ± 0.46 ^a^	0.75 ± 0.11 ^a^	0.98 ± 0.12 ^a^
WhBJ	751.35 ± 0.66 ^c^	832.86 ± 0.76 ^b^	472.16 ± 0.65 ^c^	4.66 ± 0.32 ^b^	11.21 ± 0.39 ^b^	1.60 ± 0.31 ^b^	0.87 ± 0.19 ^b^	41.31 ± 0.38 ^c^	0.71 ± 0.22 ^a^	1.19 ± 0.22 ^a^
WhRJ	515.95 ± 0.43 ^d^	771.33 ± 0.42 ^c^	410.73 ± 0.55 ^d^	3.56 ± 0.31 ^c^	5.81 ± 0.34 ^c^	0.83 ± 0.23 ^c^	1.56 ± 0.28 ^a^	45.03 ± 0.38 ^b^	0.86 ± 0.17 ^a^	1.10 ± 0.22 ^a^

The values are expressed as mean values ± standard deviations of all measurements; data sharing different letters in the same column are significantly different (*p* < 0.05), according to Duncan’s test. nd: not detectable.

**Table 4 foods-14-03193-t004:** Total polyphenol content (TPC) and antioxidant activity (AA) of raw materials and jellies fortified with whey and fruits.

Samples	Total Polyphenol Content (TPC) (mg/100 g)	Antioxidant Activity (AA), DPPH (%)
Raw Materials		
Whey	7.10 ± 0.25 ^D^	3.22 ± 0.10 ^C^
SJ	193.54 ± 0.10 ^A^	39.74 ± 0.25 ^B^
BJ	147.55 ± 0.33 ^B^	41.22 ± 0.22 ^A^
RJ	139.44 ± 0.48 ^C^	2.97 ± 0.05 ^C^
Jellies		
CJ	nd	nd
WhJ	6.79 ± 0.15 ^d^	2.45 ± 0.70 ^d^
WhSJ	196.48 ± 1.15 ^a^	25.86 ± 0.53 ^b^
WhBJ	145.32 ± 0.57 ^b^	28.11 ± 0.15 ^a^
WhRJ	136.7 ± 1.35 ^c^	3.20 ± 0.06 ^c^

The values are expressed as mean values ± standard deviations of all measurements; data sharing different letters in the same column are significantly different (*p* < 0.05), according to Duncan’s test.

**Table 5 foods-14-03193-t005:** Color analysis of jelly samples.

Sample	*L**	*a**	*b**	*C**	*h°*
CJ	46.14 ± 0.92 ^a^	–1.70 ± 0.07 ^d^	4.53 ± 0.53 ^c^	4.84 ± 0.52 ^d^	110.57 ± 1.16 ^a^
WhJ	32.8 ± 0.99 ^b^	19.5 ± 1.28 ^b^	6.66 ± 0.71 ^b^	20.61 ± 1.34 ^b^	18.86 ± 0.93 ^c^
WhBJ	18.46 ± 0.64 ^d^	5.88 ± 0.66 ^c^	7.67 ± 0.54 ^b^	9.66 ± 0.52 ^c^	52.53 ± 1.64 b
WhSJ	32.16 ± 1.08 ^b,c^	23.22 ± 1.11 ^a^	8.88 ± 0.86 ^a^	24.86 ± 1.01 ^a^	20.93 ± 1.59 ^c^
WhRJ	30.56 ± 1.11 ^c^	23.17 ± 1.02 ^a^	5.5 ± 0.4 ^c^	23.81 ± 1.01 ^a^	13.35 ± 1.64 ^d^
**Mean**	**32.02**	**14.41**	**6.65**	**16.36**	**43.65**
**SD**	**9.43**	**10.65**	**1.48**	**8.55**	**41.17**

The values are expressed as the mean ± standard deviations. Data sharing different letters in the same column are significantly different (*p* < 0.05), according to Duncan’s test.

**Table 6 foods-14-03193-t006:** Texture parameters of jelly samples.

Samples			Texture			
	Hardness (N)	Adhesiveness (mJ)	Cohesiveness (-)	Springiness Index (-)	Guminess (N)	Chewiness (mJ)
**CJ**	0.22 ± 0.01 ^b^	0.13 ± 0.06 ^c^	0.41 ± 0.03 ^a,b^	0.53 ± 0.13 ^c^	0.09 ± 0.01 ^b,c^	0.33 ± 0.15 ^c^
**WhJ**	0.27 ± 0.04 ^a^	0.33 ± 0.15 ^b,c^	0.29 ± 0.07 ^c^	0.61 ± 0.25 ^b,c^	0.08 ± 0.02 ^c^	0.46 ± 0.32 ^c^
**WhBJ**	0.32 ± 0.03 ^a^	0.66 ± 0.21 ^a^	0.47 ± 0.02 ^a,b^	0.93 ± 0.03 ^a^	0.15 ± 0.02 ^a^	1.7 ± 0.36 ^a^
**WhSJ**	0.29 ± 0.01 ^a^	0.40 ± 0.10 ^b^	0.38 ± 0.06 ^b^	0.82 ± 0.10 ^a,b^	0.11 ± 0.02 ^b^	1.03 ± 0.29 ^b^
**WhRJ**	0.31 ± 0.02 ^a^	0.40 ± 0.10 ^b^	0.49 ± 0.03 ^a^	0.88 ± 0.10 ^a^	0.15 ± 0.01 ^a^	1.7 ± 0.17 ^a^

The values are expressed as mean values ± standard deviations of all measurements. Data sharing different letters in the same column are significantly different (*p* < 0.05), according to Duncan’s test.

## Data Availability

The original contributions presented in the study are included in the article; further inquiries can be directed to the corresponding authors.

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
