# Peer review of "Whey Valorization in Functional Jellies: A Nutritional and Technological Approach"

_foods, 2025, doi:10.3390/foods14183193_

Round 1
Reviewer 1 Report
Comments and Suggestions for Authors
In this study, whey and berry juice were incorporated into cold-set gelatin gels. The physical-chemical parameters (including pH, titrable acidity, color, and textual properties), microstructure, compositions (including water content, protein content, free amino acid content, total mineral substances, macro and microelements, and total phenolic content), DPPH scavenging capacity, and storage stability of getatin jellies were determined. The results of the present work can advance current knowledge and may be beneficial to the industry. However, the scientific quality of this study is unsatisfactory. The novelty of this study is not identified in the introduction.
The objectives and the rationale of this study are clearly stated. The methods in the text were reported in sufficient detail to allow for reproducibility. The statistical reporting was not appropriate in the text. The data of the control was not indicated. The data could not support the interpretation of results and study conclusions. The data interpretation and analyses presented in the manuscript were insufficient.
The manuscript could benefit from including the results of the moisture, protein, free amino acid, macro and microelements composition, and total mineral substances content of whey and fruit juices, as well as other antioxidant activity analyses, water loss during gel formation, and CLSM for microstructure.
In a word, the quality of this study is unsatisfactory, and the manuscript is not acceptable in its present form.
Additional comments were as follows:
- Line 93-95, It was not clear how the fruit juices were prepared and the details of whey, gelatin, and sugar.
- Line 98-99, did the heating temperature destroy the color, flavor, and bioactive compounds of fruit juice?
- Line 112-113, it was unnecessary to indicate the details of calibration, which is the regular operation when using a pH meter.
- Line 114, What kinds of samples were used for pH determination? The solution, dispersion, or the solid gels?
- Line 177, how long were the samples mixed?
- Line 187-189, what was the solvent for preparing the DPPH solution? Did the turbidity and color of the samples affect the analysis?
- Line 241, were the trials repeated with different fresh samples? Please indicate it.
- Line 248-253, what was the moisture, protein, free amino acid, and total mineral substances content of whey and fruit juices? The data interpretation should consider the composition's content.
- Line 254-263, what was the water loss rate during gel formation for different samples? Water loss might lead to low moisture content and a harder texture. The statement in lines 258-260 was not correct. Please cite the literature to support the view. Moreover, the water-holding capacity of jellies was not assessed.
- Line 268-269, why? Please explain the reasons. Did the interaction result in higher protein content? Please supplement the protein content of whey and fruit juices.
- Line 290-319, the significant difference in the data should be interpreted by statistical analysis. The error bars in Figure 2 were not displayed. Please explain the reasons. Please supplement the pH and TA of whey and fruit juices.
- Line 338-375, please complement the macro and microelements compositions of whey and fruit juices. The data interpretation should consider the composition of whey and fruit juices.
- Line 384-385, what kinds of phenolic compounds were contained in whey? Please cite the literature to support the statement.
- Line 415-416, what about the error bars of the data in Figure 3?
- Line 424-425, please supplement the water-holding capacity of jellies to support the statement.
- Line 458-500, the significant difference in the data should be interpreted by statistical analysis. Was the data expression in Table 5 rational? What kinds of statistical analysis methods were used?
- The quality of microstructure images was not satisfactory. The CLSM may result in better results.
Author Response
Date: Timisoara 04.09.2025
Name: Monica Negrea
University: University of Life Sciences "King Mihai I" from Timisoara
Address:Calea Aradului No. 119, 300641 Timisoara, Romania
E-mail: monicanegrea@usvt.ro
Dear Reviewer,
We would like to address all our thanks and gratitude for the constructive observations, corrections and recommendations which contributed to the significant improvement of the paper.
Based on the reviewers’ recommendations, the authors of this paper responded point by point to the following aspects:
Reviewer 1
Comments and Suggestions for Authors
In this study, whey and berry juice were incorporated into cold-set gelatin gels. The physical-chemical parameters (including pH, titrable acidity, color, and textual properties), microstructure, compositions (including water content, protein content, free amino acid content, total mineral substances, macro and microelements, and total phenolic content), DPPH scavenging capacity, and storage stability of getatin jellies were determined. The results of the present work can advance current knowledge and may be beneficial to the industry. However, the scientific quality of this study is unsatisfactory. The novelty of this study is not identified in the introduction.
The objectives and the rationale of this study are clearly stated. The methods in the text were reported in sufficient detail to allow for reproducibility. The statistical reporting was not appropriate in the text. The data of the control was not indicated. The data could not support the interpretation of results and study conclusions. The data interpretation and analyses presented in the manuscript were insufficient.
The manuscript could benefit from including the results of the moisture, protein, free amino acid, macro and microelements composition, and total mineral substances content of whey and fruit juices, as well as other antioxidant activity analyses, water loss during gel formation, and CLSM for microstructure.
In a word, the quality of this study is unsatisfactory, and the manuscript is not acceptable in its present form.
Answer: Thank you for your comments. According to your suggestions, the paper has been significantly modified by adding data regarding control (whey and berries juice), reconsidering the results and discussions, as well as the conclusions. We hope that the modifications and additions made meet the requirements set forth by this review.
Comment: However, I have identified several shortcomings:
Comments 1: Line 93-95, It was not clear how the fruit juices were prepared and the details of whey, gelatin, and sugar.
Response 1: The information were added in the text
Comments 2: Line 98-99, did the heating temperature destroy the color, flavor, and bioactive compounds of fruit juice?
Response 2: According the literature studies, the heating fruit juice to 60–70 °C (for short pasteurization times) does not “destroy” color, flavor, or bioactives, but it can cause modest, product-dependent losses. At these mild temperatures typically inactivate spoilage enzymes and microbes while retaining most pigments, volatiles, and phenolics. Mild pasteurization (50–70 °C) can better preserve color, vitamin C, and antioxidant capacity than hotter processes and may even increase total phenolics via improved extractability.
References:
- Tchuenchieu A, Essia Ngang JJ, Servais M, Dermience M, Sado Kamdem S, Etoa FX, Sindic M. Effect of low thermal pasteurization in combination with carvacrol on color, antioxidant capacity, phenolic and vitamin C contents of fruit juices. Food Sci Nutr. 2018 Mar 6;6(4):736-746. doi: 10.1002/fsn3.611
- Oancea S. A Review of the Current Knowledge of Thermal Stability of Anthocyanins and Approaches to Their Stabilization to Heat. Antioxidants (Basel). 2021 Aug 24;10(9):1337. doi: 10.3390/antiox10091337
Comments 3: Line 112-113, it was unnecessary to indicate the details of calibration, which is the regular operation when using a pH meter.
Response 3: The correction was done, and we deleted the sentence.
Comments 4: Line 114, What kinds of samples were used for pH determination? The solution, dispersion, or the solid gels?
Response 4: We used a dispersion. The procedure was added in the text: `` 10 g from each jelly and 90 mL water were homogenize 60 s, stand 3–5 min to release air bubbles, then the pH was measured``.
Comments 5: Line 177, how long were the samples mixed?
Response 5: The samples were mixed for 30 minutes using a Holt mechanical shaker, the information was included in the text.
Comments 6: Line 187-189, what was the solvent for preparing the DPPH solution? Did the turbidity and color of the samples affect the analysis?
Response 6: The solvent used to prepare a DPPH (2,2-diphenyl-1-picrylhydrazyl) solution was ethanol. To avoid interference from the colour of the jellies with the absorption readings, the jelly samples were diluted to reduce the colour intensity, 1:10, as mentioned in the paper. In addition, prior to the absorption measurement, the samples were filtered to remove solid particles that could cause turbidity and, implicitly, errors in the spectrophotometric analysis – we completed in the paper (line 191-192).
Comments 7: Line 241, were the trials repeated with different fresh samples? Please indicate it.
Response 7: We repeated all trials on independent fresh batches and analyzed technical replicates per batch. The information was added in the text
Comments 8: Line 248-253, what was the moisture, protein, free amino acid, and total mineral substances content of whey and fruit juices? The data interpretation should consider the composition's content.
Response 8: The composition of whey and fruits juices was added in the table 2
Comments 9: Line 254-263, what was the water loss rate during gel formation for different samples? Water loss might lead to low moisture content and a harder texture. The statement in lines 258-260 was not correct. Please cite the literature to support the view. Moreover, the water-holding capacity of jellies was not assessed.
Response 9: The paragraph was reconsidered according the suggestion in this form: ``The higher moisture of the control jelly (CJ, 70.48%) is consistent with a softer, more open gel network and reduced WHC; in soft hydrogels, greater free water correlates with lower firmness and higher serum release (syneresis). Conversely, whey-based jellies (WhWJ, WhSJ, WhBJ, WhRJ; 62.3–63.2%) benefit from whey protein (α-lactalbumin, β-lactoglobulin) gelation and protein–polysaccharide interactions, which tighten the network, immobilize water, and reduce syneresis. Studies on acidified milk model gels show that added whey protein ingredients systematically decrease water mobility and syneresis and increase WHC, with microstructure–water mobility–WHC tightly correlated. Similar improvements in water retention and texture are reported for whey-containing composite gels (including gelatin–globular protein systems)``.
References
Li, R.; Czaja, T.P.; Glover, Z.J.; Ipsen, R.; Jæger, T.C.; Rovers, T.A.M.; Simonsen, A.C.; Svensson, B.; van den Berg, F.; Hougaard, A.B. Water mobility and microstructure of acidified milk model gels with added whey protein ingredients. Food Hydrocoll. 2022, 127, 107548. https://doi.org/10.1016/j.foodhyd.2022.107548
Song, D.-H.; Yang, N.-E.; Ham, Y.-K.; Kim, H.-W. Physicochemical Properties of Mixed Gelatin Gels with Soy and Whey Proteins. Gels 2024, 10, 124. https://doi.org/10.3390/gels10020124
Comments 10: Line 268-269, why? Please explain the reasons. Did the interaction result in higher protein content? Please supplement the protein content of whey and fruit juices.
Response 10: The explanation was added in the text in this form: ``The WhSJ samples displayed comparatively higher protein levels but lower free amino acid content than the other formulations. This outcome can be attributed to both compositional and interactional factors. Strawberries contribute relatively modest amounts of free amino acids compared with raspberries or blueberries, while being richer in organic acids and phenolic compounds. These phenolics, particularly ellagic acid derivatives, are prone to interact with whey proteins through non-covalent bonds, thereby stabilizing the protein matrix and limiting proteolysis or the release of small peptides and amino acids. The stronger protein–polyphenol associations may explain why WhSJ maintained higher measurable protein content but a reduced pool of free amino acids``
References:
Daly, K.; Al-Rammahi, M.; Moran, A.; Marcello, M.; Ninomiya, Y.; Shirazi-Beechey, S.P. Sensing of amino acids by the gut-expressed taste receptor T1R1–T1R3 stimulates CCK secretion. Am. J. Physiol.-Gastrointest. Liver Physiol. 2013, 304, G271–G282. https://doi.org/10.1152/ajpgi.00074.2012
Wu, H., Oliveira, G., & Lila, M. A. (2023). Protein-binding approaches for improving bioaccessibility and bioavailability of anthocyanins. Comprehensive Reviews in Food Science and Food Safety, 22, 333–354. https://doi.org/10.1111/1541-4337.13070
Enomoto H. Unique distribution of ellagitannins in ripe strawberry fruit revealed by mass spectrometry imaging. Curr Res Food Sci. 2021 Nov 17;4:821-828. doi: 10.1016/j.crfs.2021.11.006.
Comments 11: Line 290-319, the significant difference in the data should be interpreted by statistical analysis. The error bars in Figure 2 were not displayed. Please explain the reasons. Please supplement the pH and TA of whey and fruit juices.
Response 11: The figure 2 was reconsidered and the error bars were added. The pH and TA values of whey and berries juice were added in the text.
Comments 12:Line 338-375, please complement the macro and microelements compositions of whey and fruit juices. The data interpretation should consider the composition of whey and fruit juices.
Response 12: The macro and microelements compositions of whey and fruit juices was added and the results were reformulated.
Comments 13: Line 384-385, what kinds of phenolic compounds were contained in whey? Please cite the literature to support the statement.
Response 13: The information was added in the text in this form: `` Whey can contain phenolic compounds, Predominantly benzoic-acid derivatives (e.g., p-hydroxybenzoic, vanillic, syringic, salicylic) and hydroxycinnamic acids (e.g., p-coumaric, ferulic, caffeic, sinapic), but native levels are low and depend on the animal’s diet, and they increase or diversify when whey is fermented or combined with plant ingredients. In a goat study that analyzed milk, whey, and cheese by HPLC-DAD, whey showed a profile rich in benzoic-acid derivatives, while plant flavones present in feed were not found in milk/whey—evidence that simple phenolic acids are the dominant transferrable species``.
References: Sepe, Lucia & Cornu, Agnes & Graulet, Benoit & Salvatore, Claps & Rufrano, Domenico. (2011). Phenolic content of forage, milk, whey and cheese from goats fed Avena sativa.
Comments 14: Line 415-416, what about the error bars of the data in Figure 3?
Response 14: The errors bars were included in the graph presented in figure 3
Comments 15: Line 424-425, please supplement the water-holding capacity of jellies to support the statement.
Response 15: The information was added in this form: ``Lower syneresis at 4 °C is consistent with a tighter, colder gel network that restricts water mobility; by contrast, storage at 20–25 °C increases molecular mobility and promotes serum release. In mixed protein–hydrocolloid gels, water-holding capacity (WHC) and syneresis are inversely related and track the gel’s microstructure (pore size/rigidity) and bound vs. free water fractions. Studies on dairy/acidified milk model systems show that added whey proteins decrease water mobility and improve WHC, and that microstructural indices (e.g., T₂ relaxation) correlate with lower syneresis during storage``.
References: Li, R.; Czaja, T.P.; Glover, Z.J.; Ipsen, R.; Jæger, T.C.; Rovers, T.A.M.; Simonsen, A.C.; Svensson, B.; van den Berg, F.; Hougaard, A.B. Water mobility and microstructure of acidified milk model gels with added whey protein ingredients. Food Hydrocoll. 2022, 127, 107548. https://doi.org/10.1016/j.foodhyd.2022.107548
Song, D.-H.; Yang, N.-E.; Ham, Y.-K.; Kim, H.-W. Physicochemical Properties of Mixed Gelatin Gels with Soy and Whey Proteins. Gels 2024, 10, 124. https://doi.org/10.3390/gels10020124
Comments 16: Line 458-500, the significant difference in the data should be interpreted by statistical analysis. Was the data expression in Table 5 rational? What kinds of statistical analysis methods were used?
Response 16: The results presented in the table 5 were interpreted from statistical point of view and the information were added in the table. The Duncan test was used and included in the explanation
Comments 17: The quality of microstructure images was not satisfactory. The CLSM may result in better results.
Response 17: The images were improved in 1280 pixels and 300 DPI
Once again, we would like to thank the reviewer for your appreciations, corrections and recommendations which contributed to the significant improvement of the paper.

Reviewer 2 Report
Comments and Suggestions for Authors
This study utilizes whey combined with fruit juice to develop a novel health-focused jelly, analyzed its composition and functional parameters. Some revisions are advised as following:
- The preparation method offruit juices is not specified (e.g., the concentrate of fruit juices).
- The type and composition of cow's milkwhey need to be completed.
- The sensory change was mentioned, but the specific sensory evaluation method (such as scoring criteria, qualifications of evaluators) and data were not provided in the study, please check it and supply the related method and data.
- Page 14, line 567, “coresponds”is misspelled, it should be corresponds.
- Page 2, line 100, and elsewhere in the text, “gelatin”or “gelatine”should be consistent.
- On page 3, line 101, the word "~" can be changed to "approximately" or "around" to avoid the use of informal symbols in academic writing.
- The word "amino acids" appears many times in the text, which should be changed to the fixed collocation "amino acids";
- Page 6, line 260, according to the context, WhBJ should be "whey blueberry jelly",is here "blackberry" a spelling error?
- Standard deviation should be added following the average in Figure 3 and table 5 .
Author Response
Date: Timisoara 04.09.2025
Name: Monica Negrea
University: University of Life Sciences "King Mihai I" from Timisoara
Address:Calea Aradului No. 119, 300641 Timisoara, Romania
E-mail: monicanegrea@usvt.ro
Dear Reviewer,
We would like to address all our thanks and gratitude for the constructive observations, corrections and recommendations which contributed to the significant improvement of the paper.
Based on the reviewers’ recommendations, the authors of this paper responded point by point to the following aspects:
Comments:
This study utilizes whey combined with fruit juice to develop a novel health-focused jelly, analyzed its composition and functional parameters. Some revisions are advised as following:
Comments 1: The preparation method of fruit juices is not specified (e.g., the concentrate of fruit juices).
Response 1: The information was added in the methodology in this form: „To obtain 300 ml of juice, 700-800 g of fresh fruit was used, which was cleaned, washed, and blended using a mixer (Braun TriForce Power Blend 9, Braun, Neu-Isenburg, Germany), after which the resulting product was strained to obtain the liquid sample„
Comments 2: The type and composition of cow's milkwhey need to be completed.
Response 2: The whey used was sweet whey derived from cheese production and the composition was added to the methods
Comments 3: The sensory change was mentioned, but the specific sensory evaluation method (such as scoring criteria, qualifications of evaluators) and data were not provided in the study, please check it and supply the related method and data.
Response 3: The sensory analysis based on hedonic scale and using panel of consumers was not presented in this study, the sensory changes refer only at visual aspects after different days of storage in different conditions. The correction was done in the text
Comments 4: Reviewer: Page 14, line 567, “coresponds”is misspelled, it should be corresponds.
Response 4: The correction has been made.
Comments 5: Page 2, line 100, and elsewhere in the text, “gelatin”or “gelatine”should be consistent.
Response 5 : The word "gelatin" (American English) was replaced with "gelatine" (British English) throughout the entire text
Comments 6: On page 3, line 101, the word "~" can be changed to "approximately" or "around" to avoid the use of informal symbols in academic writing.
Response 6: The correction has been made.
Comments 7: The word "amino acids" appears many times in the text, which should be changed to the fixed collocation "amino acids";
Response 7: The correction has been made.
Comments 8: Page 6, line 260, according to the context, WhBJ should be "whey blueberry jelly",is here "blackberry" a spelling error?
Response 8: Yes! The correction has been made.
Comments 9: Standard deviation should be added following the average in Figure 3 and table 5 .
Response 9: Standard deviation was added for figure 3 and table 5
Once again, we would like to thank the reviewer for your appreciations, corrections and recommendations which contributed to the significant improvement of the paper.

Reviewer 3 Report
Comments and Suggestions for Authors This article presents a wealth of research findings, is underpinned by robust data, and is built upon a well-structured framework, thereby conferring substantial academic value and practical relevance. Nevertheless, there remains room for further refinement.- It is recommended that the experimental conditions employed for the assessment of antioxidant activity—such as concentrations and incubation times—be reported in full detail.
- Several subsection numbers are duplicated (e.g., 2.5.1. and 2.5.). These should be systematically reviewed and formatted for consistency.
- In Table 5 (color parameters), the inclusion of significance indicators (e.g., superscript letters) is advised.
- The conclusion could be strengthened by more explicitly emphasizing the study’s innovative contributions and its broader implications, particularly with respect to advancing the circular economy and mitigating food waste.
- The clarity of numerous figures should be enhanced.
Author Response
Date: Timisoara 04.09.2025
Name: Monica Negrea
University: University of Life Sciences "King Mihai I" from Timisoara
Address:Calea Aradului No. 119, 300641 Timisoara, Romania
E-mail: monicanegrea@usvt.ro
Dear Reviewer,
We would like to address all our thanks and gratitude for the constructive observations, corrections and recommendations which contributed to the significant improvement of the paper.
Based on the reviewers’ recommendations, the authors of this paper responded point by point to the following aspects:
Comment
This article presents a wealth of research findings, is underpinned by robust data, and is built upon a well-structured framework, thereby conferring substantial academic value and practical relevance. Nevertheless, there remains room for further refinement.
Comments 1: It is recommended that the experimental conditions employed for the assessment of antioxidant activity—such as concentrations and incubation times—be reported in full detail.
Response 1: The experimental part regarding antioxidant activity was reformulated and suggested information were added.
Comments 2: Several subsection numbers are duplicated (e.g., 2.5.1. and 2.5.). These should be systematically reviewed and formatted for consistency.
Response 2: Section 2.5 is correctly divided into two subsections: 2.5.1 and 2.5.2. There is no duplication; the confusion was due to a missing space in formatting. The spacing has now been corrected for consistency.
Comments 3: In Table 5 (color parameters), the inclusion of significance indicators (e.g., superscript letters) is advised.
Response 3: The statistics parameters were added to the table 5
Comments 4: The conclusion could be strengthened by more explicitly emphasizing the study’s innovative contributions and its broader implications, particularly with respect to advancing the circular economy and mitigating food waste.
Response 4: The conclusion was reformulated and aspects regarding innovative contribution and the importance of whey in the circular economy was highlighted
Comments 5: The clarity of numerous figures should be enhanced.
Response 5: The quality of the figures was improved.
Once again, we would like to thank the reviewer for your appreciations, corrections and recommendations which contributed to the significant improvement of the paper.

Reviewer 4 Report
Comments and Suggestions for Authors
The manuscript discusses the application of whey and different berries for formulation of fruit-based jellies. The authors assessed the physicochemical and sensorial properties of the jellies. The topic of the work should be of interest to the readership of the journal. However, the manuscript should be improved by an inclusion of additional discussions and information in some sections. Several existing grammatical errors should also be corrected.
I have the following comments/suggestions to the authors:
- In the keywords section, I suggest adding the words “functional jellies” and “whey” to the list of keywords. The keywords “blueberries”, “strawberries”, and “raspberries” can simply be presented as “berries”.
- In Figure 1, please check if the images for control jelly and whey jelly are accurately presented in figures 1a and b or the figures should be swapped. The lightness of samples are not in accordance with the values of presented L* currently.
- In section 2.3.4, please mention in the main text the absorption was measured at which wavelength.
- Referring to lines 268-269, provide discussions on how WhSJ showed higher protein and lower free amino acid content compared with the other jellies.
- Referring to lines 255-257, polyphenols can form complexes with proteins. Please explain and add additional discussions to the main text on how free amino acids can be available and positively influence protein digestion.
- Referring to lines 369-371, Ni or Cr are not reported to provide health benefits compared to other minerals reported in the reference. Please provide adequate discussion on this statement.
- In line 392, please provide the complete term for the abbreviation “FW”.
- Referring to lines 394-395, provide examples of polyphenols in raspberries that did not show antioxidant effect.
- Several existing grammatical errors or typing mistakes should be corrected:
- Line 40, “it can be conclude”
- In line 58, “lellies”
- In line 82, “respond”
- In line 127, “2 different temperature”
- In line 198, “The releases liquid”
- In line 257, “WhWJ”
Comments on the Quality of English Language
The existing grammatical errors should be corrected.
Author Response
Date: Timisoara 04.09.2025
Name: Monica Negrea
University: University of Life Sciences "King Mihai I" from Timisoara
Address:Calea Aradului No. 119, 300641 Timisoara, Romania
E-mail: monicanegrea@usvt.ro
Dear Reviewer,
We would like to address all our thanks and gratitude for the constructive observations, corrections and recommendations which contributed to the significant improvement of the paper.
Based on the reviewers’ recommendations, the authors of this paper responded point by point to the following aspects:
Comments
The manuscript discusses the application of whey and different berries for formulation of fruit-based jellies. The authors assessed the physicochemical and sensorial properties of the jellies. The topic of the work should be of interest to the readership of the journal. However, the manuscript should be improved by an inclusion of additional discussions and information in some sections. Several existing grammatical errors should also be corrected.
I have the following comments/suggestions to the authors:
Comments 1: In the keywords section, I suggest adding the words “functional jellies” and “whey” to the list of keywords. The keywords “blueberries”, “strawberries”, and “raspberries” can simply be presented as “berries”.
Response 1: The corrections have been made.
Comments 2: In Figure 1, please check if the images for control jelly and whey jelly are accurately presented in figures 1a and b or the figures should be swapped. The lightness of samples are not in accordance with the values of presented L* currently.
Response 2: Thank you for the report, indeed, the figures were mistakenly reversed, the mistake has been corrected.
Comments 3: In section 2.3.4, please mention in the main text the absorption was measured at which wavelength.
Response 3: The used wavelength was 570 nm. The information was added in the text.
Comments 4: Referring to lines 268-269, provide discussions on how WhSJ showed higher protein and lower free amino acid content compared with the other jellies.
Response 4: The information was added in the text in this form: „The WhSJ samples (whey + strawberry jellies) displayed comparatively higher protein levels but lower free amino acid content than the other formulations. This outcome can be attributed to both compositional and interactional factors. Strawberries contribute relatively modest amounts of free amino acids compared with raspberries or blueberries, while being richer in organic acids and phenolic compounds. These phenolics, particularly ellagic acid derivatives, are prone to interact with whey proteins through non-covalent bonds, thereby stabilizing the protein matrix and limiting proteolysis or the release of small peptides and amino acids. The stronger protein–polyphenol associations may explain why WhSJ maintained higher measurable protein content but a reduced pool of free amino acids. From a technological perspective, this balance influences the textural and sensory properties of the jellies. Higher protein content can support a firmer gel network, con-tributing to structural stability, while the reduced free amino acid content may moderate flavor development compared with berry jellies richer in amino acid precursors. Nutritionally, the retention of intact protein fractions in WhSJ highlights the capacity of strawberry-derived compounds to protect whey proteins against degradation, though it may limit the immediate availability of free amino acids, which are more rapidly absorbed in the human body„.
References: Daly, K.; Al-Rammahi, M.; Moran, A.; Marcello, M.; Ninomiya, Y.; Shirazi-Beechey, S.P. Sensing of amino acids by the gut-expressed taste receptor T1R1–T1R3 stimulates CCK secretion. Am. J. Physiol.-Gastrointest. Liver Physiol. 2013, 304, G271–G282. https://doi.org/10.1152/ajpgi.00074.2012
Wu, H., Oliveira, G., & Lila, M. A. (2023). Protein-binding approaches for improving bioaccessibility and bioavailability of anthocyanins. Comprehensive Reviews in Food Science and Food Safety, 22, 333–354. https://doi.org/10.1111/1541-4337.13070
Enomoto H. Unique distribution of ellagitannins in ripe strawberry fruit revealed by mass spectrometry imaging. Curr Res Food Sci. 2021 Nov 17;4:821-828. doi: 10.1016/j.crfs.2021.11.006.
Comments 5: Referring to lines 255-257, polyphenols can form complexes with proteins. Please explain and add additional discussions to the main text on how free amino acids can be available and positively influence protein digestion.
Response 5: The expalanation was added in the text in this form: „Polyphenols, such as anthocyanins, flavonols, and phenolic acids abundant in berries, are known to interact with proteins through hydrogen bonding, hydrophobic interactions, and, in some cases, covalent linkages. These interactions can lead to the formation of protein–polyphenol complexes, which influence solubility, digestibility, and the release of bioactive compounds. In the case of whey-based jellies, the extent of complex formation helps explain differences in protein and free amino acid content across formulations. Stronger interactions may preserve protein integrity, while weaker interactions or enzymatic activity during processing can release small peptides and free amino acids.
From a digestive standpoint, the coexistence of intact whey proteins, peptides, and free amino acids in the same food matrix can be considered advantageous. While intact proteins ensure sustained amino acid release during digestion, free amino acids provide an immediate nutritional boost, balancing rapid absorption with prolonged availability. Therefore, differences in free amino acid concentrations among the whey–berry jellies are not only a marker of protein–polyphenol interactions but also have nutritional significance by influencing the kinetics of protein digestion and amino acid uptake„.
References:
Feng, Y.; Jin, C.; Lv, S.; Zhang, H.; Ren, F.; Wang, J. Molecular Mechanisms and Applications of Polyphenol-Protein Complexes with Antioxidant Properties: A Review. Antioxidants 2023, 12, 1577. https://doi.org/10.3390/antiox12081577
Chima, B.; Mathews, P.; Morgan, S.; Johnson, S.A.; Van Buiten, C.B. Physicochemical Characterization of Interactions between Blueberry Polyphenols and Food Proteins from Dairy and Plant Sources. Foods 2022, 11, 2846. https://doi.org/10.3390/foods11182846.
Comments 6: Referring to lines 369-371, Ni or Cr are not reported to provide health benefits compared to other minerals reported in the reference. Please provide adequate discussion on this statement.
Response 6: The discussions were added in the text in this form: „The control jelly (CJ) showed intermediate Ni content, suggesting minor background from non-berry ingredients or matrix effects, whereas adding whey alone did not in-crease Ni content (WhJ). Cr was not detected in CJ and WhJ but was consistently detected—and at comparable magnitudes—in all berry-containing jellies, indicating that berries are the principal source of both Ni and Cr in these formulations. This aligns with known trace-metal uptake by small fruits from soil and irrigation water; by contrast, the absence of Cr in WhJ argues against appreciable process-equipment leaching under the applied conditions.
From a safety/quality standpoint, the whey does not elevate Ni/Cr, while the berry component dictates trace-metal presence; therefore, raw-material sourcing and agro-nomic provenance (soil, cultivar, farming practice) are the key levers to manage Ni/Cr levels ``.
Reference: Dhalaria, R.; Verma, R.; Kumar, D.; Puri, S.; Tapwal, A.; Kumar, V.; Nepovimova, E.; Kuca, K. Bioactive Compounds of Edible Fruits with Their Anti-Aging Properties: A Comprehensive Review to Prolong Human Life. Antioxidants 2020, 9, 1123. https://doi.org/10.3390/antiox9111123.
Comments 7: In line 392, please provide the complete term for the abbreviation “FW”.
Response 7: FW is fresh weight, we completed in the text the complete term.
Comments 8: Referring to lines 394-395, provide examples of polyphenols in raspberries that did not show antioxidant effect.
Response 8: The information was added in the text in this form: ``While most raspberry anthocyanins are antioxidant, glycosylation pattern can reduce activity substantially (e.g., certain 3-glycosides/arabinosides show markedly lower activity vs. aglycones in DPPH/ABTS/FRAP), meaning little or undetectable effect at typical test concentrations depending on the assay/design. Also, native raspberry ellagitannins are potent antioxidants in vitro, but their bioavailable forms (urolithins, esp. Uro-B and methylated urolithins) often lack direct radical-scavenging in standard chemical assays``
References: Sadowska-Bartosz I, Bartosz G. Antioxidant Activity of Anthocyanins and Anthocyanidins: A Critical Review. Int J Mol Sci. 2024 Nov 8;25(22):12001. doi: 10.3390/ijms252212001.
Banc, R.; Rusu, M.E.; Filip, L.; Popa, D.-S. The Impact of Ellagitannins and Their Metabolites through Gut Microbiome on the Gut Health and Brain Wellness within the Gut–Brain Axis. Foods 2023, 12, 270. https://doi.org/10.3390/foods12020270
Comments 9: Several existing grammatical errors or typing mistakes should be corrected:
- Line 40, “it can be conclude
- In line 58, “lellies”
- In line 82, “respond”
- In line 127, “2 different temperature”
- In line 198, “The releases liquid”
- In line 257, “WhWJ”
Response 9: The corrections have been made.
Once again, we would like to thank the reviewer for your appreciations, corrections and recommendations which contributed to the significant improvement of the paper.

Reviewer 5 Report
Comments and Suggestions for Authors
see attached file

Author Response
Date: Timisoara 04.09.2025
Name: Monica Negrea
University: University of Life Sciences "King Mihai I" from Timisoara
Address:Calea Aradului No. 119, 300641 Timisoara, Romania
E-mail: monicanegrea@usvt.ro
Dear Reviewer,
We would like to address all our thanks and gratitude for the constructive observations, corrections and recommendations which contributed to the significant improvement of the paper.
Based on the reviewers’ recommendations, the authors of this paper responded point by point to the following aspects:
Comment:
The concept of the study is straightforward, but its presentation in the manuscript needs to be seriously improved.
Comment 1. The overall design of the presentation of the results and especially their discussion seems confusing. Five types of jellies were studied: 1. Gelatin; 2. Whey Jelly, in which Whey was added to gelatin; 3, 4, and 5 – Whey Strawberry, Whey Raspberry, and Whey Blueberry Jellies. The nature and chemical composition of the ingredients make it obvious that Whey Jelly (2) should be compared to gelatin jelly (1), while jelly 3, 4, and 5 should be compared to Whey Jelly without berry juice (2). With such a comparison, the authors will be able to clearly answer the question of how whey improves on gelatin jelly, as well as the question of how berry juice improves whey jelly. Comparing and discussing whey jelly enriched with berries with gelatin jelly is incorrect and insignificant.
Response 1: The results and discussions were reinterpreted according to the requirements, specifically the values obtained for the physicochemical parameters of control samples (1) were compared with the values obtained for whey gels (sample 2), while the samples with added beeries juice (3, 4, 5) were compared with sample 2 with added whey.
Comment 2. It is strange that the manuscript does not contain data on the composition and properties of berry juices (protein, free amino acids, mineral substances, macro/microelement composition, total polyphenol content, and antioxidant activity). Obviously, the differences in the parameters indicated between the jelly samples are due to differences in the composition of juices obtained from different berries. It is recommended to provide the composition of the berry juices used.
Response 2: The experimental data have been supplemented with the values obtained for whey powder and berries juice for the analyzed physicochemical parameters, and the interpretations of the results have been reformulated according to the requirements.
Comment 3. The photographs in section 3.8 are of very low quality. Their description in the text does not correspond at all to what the reader sees in the photograph. For example, such characteristics as homogeneity, density, and porosity are not recognizable in the images. It is recommended that they be removed from the manuscript or replaced with others.
Response 3: The images were improved 1280 pixels and 300 dpi
Comment 4. Section 3.9 should be removed from the manuscript, since conducting a correlation analysis with n=3 in each group is incorrect and may lead to erroneous conclusions.
Response 4: Thank you for your feedback. We understand your concern regarding the low sample size (n=3) for the correlation analysis in Section 3.9. While a small sample size can indeed introduce uncertainty and may lead to erroneous conclusions, we believe that removing this section would be a loss to the manuscript.
The reason for our confidence in these results, despite the small sample size, is twofold. First, the variability within each group's measurements is very low, indicating a high degree of consistency and reproducibility. The data points are tightly clustered, which strengthens our ability to draw meaningful conclusions even with limited observations.
Second, the correlation analysis was not performed on the individual measurements themselves, but on the mean values derived from these consistent measurements. This approach leverages the stability of the data to provide a reliable basis for our analysis.
We believe that this section, along with the described limitations, provides valuable insight into the relationship between the variables. We are confident that the findings, while preliminary, are well-supported by the consistency of our data. We would prefer to keep this section and add a clear statement addressing the low sample size as a limitation, ensuring the reader is fully aware of the context.
Comment 5. Everywhere in the text it is necessary to remove statements about the sensory value of the obtained jellies. They were not determined in the study; therefore, they remain unknown for these samples.
Response 5: Thank you for the observation. The statements regarding sensory analysis was removed from the text. This attribute is kept in the text only when referring to texture which was analysed.
Comment 6. The following statements in paragraph Lines 321-324 are unsubstantiated and must be removed or substantiated: «risks becoming organoleptically unpleasant», «the sensory and microbiological stability is compromised», «organoleptic changes (color, flavor) are recorded, but not as obvious as in whey/fruit jellies».
Response 6: The paragraph was reformulated.
Comment 7. Figure 2: Standard deviation and significance of differences between groups must be included.
Response 7: Figure 2 was redesigned
Comment 8. It is recommended to rework Figure 3 into Figure 3A (4°C) and Figure 3B (20-25°C). In both cases, it is better to use a histogram. It is also necessary to indicate the standard deviation and the reliability of differences between groups.
Response 8: The recomandation was applied and 2 figures are presented. The sd were included to the graphs
Comment 9. The meaning of the following sentences is unclear or questionable. Try rewriting them: Lines 268-270: In particular, strawberry jellies (WhSJ) presented the highest protein level, possibly due to the interactions between whey proteins and bioactive compounds in the fruits.
Lines 273-275: Raspberry jellies (WhRJ) recorded the highest concentration of free amino acids, suggesting a more efficient release of protein compounds under enzymatic action and bioactive compounds from raspberries
Line 301: … due to the residual activity of lactic acid bacteria. (?)
Line 345: sweet or sour whey. (?)
Lines 364-365: This distribution indicates a possible synergistic interaction between whey and blueberries
Lines 547-548: … with a positive impact on the perceived quality of the final product.
Response 9: These paragraphs were reformulated, deleted or corrected.
Comment 10. In the discussion for section 3.6 Texture analysis of whey jellies, it is necessary to explain how the type of berry juice does not affect hardness but changes other textural parameters. What molecular interaction could explain this?
Response 10: The discussion were added to the text: `` When the primary network (gelatin + whey proteins) and the total solids are con-trolled, the type of berry juice acts as a fine modulator of cohesiveness, elasticity, and adhesion through protein–polyphenol interactions and environmental effects (pH/ions/pectin/water), while the hardness remains dominated by the gel base and therefore does not significantly differ between fruits.Berry fruit juices bring pH, ions (K⁺, Ca²⁺), and small amounts of pectin that alter the charge and electrostatic screening in the network, affecting elastic recovery (springiness) and cohesion, without necessarily altering maximum compressive strength. The literature shows that moderate variations in polysaccharide/ionic composition can shift brittleness and adhesiveness, leaving 'hardness' almost constant when the gel base (gelatin/protein) is identical``.
References:
Spahn, G.; Baeza, R.; Santiago, L.G.; Pilosof, A.M.R. Whey Protein Concentrate/λ-Carrageenan Systems: Effect of Processing Parameters on the Dynamics of Gelation and Gel Properties. Food Hydrocoll. 2008, 22, 1504–1512. https://doi.org/10.1016/j.foodhyd.2007.10.002
Said, N.S.; Olawuyi, I.F.; Lee, W.Y. Pectin Hydrogels: Gel-Forming Behaviors, Mechanisms, and Food Applications. Gels 2023, 9, 732. https://doi.org/10.3390/gels9090732
Casas-Forero, N.; Orellana-Palma, P.; Petzold, G. Comparative Study of the Structural Properties, Color, Bioactive Compounds Content and Antioxidant Capacity of Aerated Gelatin Gels Enriched with Cryoconcentrated Blueberry Juice during Storage. Polymers 2020, 12, 2769. https://doi.org/10.3390/polym12122769
Comment 11. It is not clear why the term "firmness" is sometimes used instead of "hardness"?
Response 11: The term ``firmness`` was replace with ``hardness`` in all document
Comment 12. The conclusion needs to be rewritten. The current version is full of empty statements. For example, what does a "functional" product mean? Why is the jellies' nutritional value regarded as "high" rather than, say, "average"? At what values does the nutritional value become "high"? What "sensory changes" are we talking about if no sensory analysis was performed? What does the "most favorable" microstructure mean? The last paragraph of the conclusion merely duplicates the motivation from the introduction that initiated the study.
Response 12: The conclusion was reformulated
Once again, we would like to thank the reviewer for your appreciations, corrections and recommendations which contributed to the significant improvement of the paper.

Round 2
Reviewer 1 Report
Comments and Suggestions for Authors
The authors have responded to the comments.
Reviewer 4 Report
Comments and Suggestions for Authors
My comments to the authors are addressed and I have no additional major comment.
Still, minor existing typing mistakes should be corrected by the authors.
- In line 108, “arround”
- In Table 2, ensure that the value of 1230.78 for CJ is accurately presented.
- Line 508, “phyrochemical”, “beeries”
The existing typing mistakes should be corrected.
Reviewer 5 Report
Comments and Suggestions for Authors
The authors answered my questions and improved the manuscript in accordance with my comments.
Although I still have doubts about the necessity and correctness of sections 3.8 and 3.9, I consider it acceptable to publish this manuscript.